

# Hydrodynamics of long-duration urban floods: experiments and numerical modelling

Anaïs Arrault[1,3], Pascal Finaud-Guyot[2], Pierre Archambeau[1], Martin Bruwier[1], Sébastien Erpicum[1], Michel Pirotton[1] & Benjamin Dewals[1]

[1] Research group HECE, Department ArGEnCo, University of Liege, Liege, Belgium
[2] ICube, Université de Strasbourg, CNRS (UMR 7357), ENGEES, 2 rue Boussingault, Strasbourg, France
[3] Ecole des Mines d'Ales

*Correspondence to*: B. Dewals (b.dewals@ulg.ac.be)

**Abstract.** Flood risk in urbanized areas raises increasing concerns as a result of demographic and climate changes. Hydraulic modelling is a key component of urban flood risk analysis. Yet, detailed validation data are still lacking for comprehensively validating hydraulic modelling of inundation flow in urbanized floodplains. In this study, we present an experimental model of inundation flow in a typical European urban district and we compare the experimental observations with predictions by a shallow-water numerical model. The setup is 5 m × 5 m and involves seven streets along each direction, leading to 49 intersections. Different inflow discharges and flow partitions were tested. The performance of the numerical model is assessed and the upscaling of the experimental observations to the field is discussed.

## 1    Introduction

Floods are the most common natural hazard. In Europe, they caused around 100 billion euros of damage between 1986 and 2006 (De Moel et al., 2009). Many cities were built along rivers, deltas or in coastal areas and are therefore partly located in the floodplains. Urban flood risk management is of particularly high relevance as urbanization is growing at an unprecedented pace and hydrological extremes tend to increase in magnitude and frequency (Domeneghetti et al., 2015; Merz et al., 2012; Vorogushyn & Merz, 2013). Reliable predictions of flood hazard are a prerequisite to support contemporary flood risk management policies. This includes the accurate estimation of inundation extents, water depths, discharge partition and flow velocity in urbanized floodplains, since these parameters are critical inputs for flood impact modelling (Brazdova & Riha, 2014; Kellermann et al., 2015; Kreibich et al., 2014).

Models based on the shallow-water equations are the most common approach for detailed inundation modelling (Costabile & Macchione, 2015). They are considered as state-of-the-art for large scale real-world applications and were the focus of much research over the last two decades. They also benefit from widely available detailed datasets obtained from remote sensing techniques, such as laser altimetry (Dottori et al., 2013).

Different state-of-the-art hydraulic models exist to compute the inundation characteristics. However, their validation for urban flood configurations remains incomplete as reference data from the field are relatively scarce and difficult to obtain





(Dottori et al., 2013; El Kadi Abderrezzak et al., 2009; Neal et al., 2009). Water marks and aerial imagery provide some relevant information but they are far from sufficient to reflect the whole complexity of inundation flows in densely urbanized floodplains.

To address this lack of validation data, scale model studies are particularly valuable, since they deliver accurate measurements of flow characteristics under controlled hydraulic conditions (e.g., known distribution of inflow discharge and downstream boundary conditions). Several experimental studies of relevance were conducted, mainly during the last decade. They involved different degrees of complexity and realism in the considered flow configurations, ranging from isolated street intersections up to complete urban districts with surface and underground flow.

The flow behaviour in intersections is an essential component of urban flooding since a urban district may be seen as a network of streets and intersections. A limited number of experimental studies investigated specifically the flows at street intersections.

- Weber et al. (2001) presented detailed 3D flow and turbulence measurements at a 90° open-channel junction.

- Mignot et al. (2008) studied a junction of four branches, with supercritical inflow from two branches. They identified three different flow regimes, depending on the location of hydraulic jumps, either normal to the flow in the upstream channels or oblique within the junction.

- Rivière et al. (2011) studied subcritical flow in a four branch open-channel intersection. Based on 220 experiments, an empirical correlation was found for the flow partition as a function of the inflow discharges and the height of the downstream weirs.

- El Kadi Abderrezzak et al. (2011) and Rivière et al. (2014) studied transcritical flow at three and four branch intersections. They identified up to four flow regimes and showed the appearance of a critical section and a recirculation in the lateral channel. Unlike previously assumed, they also highlighted that the existence of this critical section is not sufficient to ascertain that flow in the intersection is decoupled from downstream flow conditions. The ratio between the outflow discharge in the lateral channel and the inflow discharge was correlated to the Froude number and the ratio between upstream critical depth and width of the main channel. However, all these studies were conducted for a single intersection only and considering right angles as well as channels of equal width.

- Mignot et al. (2013) studied the flow in a 3-branch bifurcation with one or several obstacles. The obstacles were 5×5 cm impervious blocks with a height of 20 cm. They cause pile up of water immediately upstream. The discharge partition was measured for nine obstacle configurations. The obstacles have different impacts on the flow depending on their location. If they are in the upstream branch they reduce the flow in the lateral branch and increase the downstream discharge. The obstacles located in the downstream branch block the flow in this branch and increase the discharge of the lateral branch. Obstacles in the lateral branch can have different behaviours.



Instead of focusing on a single street intersection, other experimental studies analysed the flow field in a whole urban district. They are however very rare and we could only identify four significant contributions.

- Zech & Soares-Frazão (2007) investigated transient flow in an idealized urban district located on a scale model of the Toce valley in Italy. Two different layouts of 20 building blocks of 15 cm × 15 cm were considered (aligned vs staggered). Water depths were measured by electrical conductivity gauges at some locations around and within the urban district. However, they do not provide a truly distributed view of the flow pattern over the whole urban district. Considering a simplified geometric setting with a flat topography, similar tests were conducted by Soares-Frazão & Zech (2008) under transient flow conditions in an experimental flume. The idealized urban district was made of 5 × 5 building blocks, either aligned or inclined with respect to the main flow direction. Water levels were measured at approximately one hundred points. Large scale particle image velocimetry (LS-PIV) was used to estimate the surface velocity field. For these two models with transient flow, boundary conditions were difficult to manage and the discharge partition between the different streets was not investigated at all.

- Ishigaki et al. (2003) used a scale model of a 1 km × 2 km district of Kyoto to highlight the importance of underground space (parking lots, subway platforms…) during moderate floods of limited duration. Water depths were measured in eight locations using ultrasonic sensors, while the surface velocity field was measured by LS-PIV. About 50% of the inflow volume was found to reach the underground space.

- A scale model of 17[th] Street of New Orleans was used by Sattar et al. (2008) to study the urban flooding induced by a dike breach during hurricane Katrina in 2005. The model was designed to evaluate the viability of using sandbags in a multibarrier configuration to effectively close the breach. The flow in the urbanized floodplain was represented to properly reflect its influence on the tailwater at the breach location. Water depths were measured at about 800 points using mechanical gauges and flow velocity was measured by a micro acoustic Doppler velocimeter, at a limited number of grid points, mainly outside the urbanized area. The outflow discharge was measured in each street.

- Lipeme Kouyi et al. (2010) describe experiments in an idealized urban district. The district consists of seven streets aligned from north to south, crossing seven other streets aligned from west to east. Most streets are not straight, so that complex geometric configurations arise at the intersections. The scale factor is 100 horizontally and 25 vertically. Only steady flows were considered and the setup did not enable direct control of the partition of the inflow discharge between the different streets. Walls were positioned west and east of the district, so that only a single main flow direction could be analysed.

In the present study, we consider the Icube Laboratory scale model of a typical European district, similar to the configuration considered by Lipeme Kouyi et al. (2010). However, here, the boundary conditions allow for the flow distribution to be varied continuously between west-east and north-south directions. The present setup enables the inflow discharge to be controlled independently in each street. We focus on steady flow conditions, which constitute a realistic approximation for long-duration floods. Flash floods are therefore out of the scope of the present study.



The study provides new insights into the distribution of water depths and the discharge partition in a more general urban setting than considered in previous research. The experimental dataset is available for testing numerical models and we provide here a comparison with the results of one specific model, namely WOLF 2D, which has been widely used for inundation mapping (Erpicum et al., 2010a) and in flood risk analysis research (Beckers et al., 2013; Bruwier et al., 2015;
Detrembleur et al., 2015; Ernst et al., 2010).

The experimental and numerical models are presented in Sect. 2. The influence of varying the total inflow discharge is analysed in Sect. 3, together with a sensitivity analysis of the computed results with respect to the main modelling characteristics. Section 4 details how the partition of the inflow discharge influences the results. In Sect. 5, the upscaling of the experimental findings to real-world applications is discussed, as well as the enhancements brought by an improved
modelling of the streets geometry. Conclusions and perspectives are given in Sect. 6.

## 2   Methods

### 2.1   Experiments

The experiments considered here were conducted by Araud (2012) at the laboratory ICube in Strasbourg (France). Compared to previous studies, the experimental setup achieves a relatively high degree of realism by involving streets of various widths
and intersections of different types (both normal branches and branches of different inclinations). The study focused on *long-duration* and *extreme* events, i.e. a steady state was considered and the flow through the underground networks was assumed negligible and was not reproduced in the model. Only an overview of the experimental setup and procedure is given here, while all details were described by Araud (2012).

#### 2.1.1   Laboratory setup

The experimental setup represents an idealized urban district. It extends over 5 m by 5 m and contains 64 impervious blocks in Plexiglass (Figure 1). These blocks define a total of 14 streets. Seven of them (noted 1 to 7) are aligned along the east-west direction, while seven other streets (noted A to G) follow the north-south direction. All streets have a width of 5 cm, except streets 4, C and F which are 12.5 cm wide. The coordinates of the geometry of the model are provided in Araud (2012). They correspond to the "as-built" coordinates of the obstacles.
The street inlets are located along the north and west faces of the model, while the outlets are on the south and east faces. Fourteen pumps were used to control individually the inflow discharge into each street. The discharge was distributed between the streets of each face proportionally to their widths. The model was fed with water, assuming no sediment transport and no debris in the flow. The outlets enable free flow conditions.





### 2.1.2 Instrumentation

The outflow discharge in each street was determined from the rating curve of calibrated weirs located downstream of the street outlets. An ultrasound sensor was used for measuring the water level upstream of each weir. With a measurement window of minimum 40 s, the uncertainty on the outflow discharge was shown to remain below 3.5 % (Araud, 2012). Given the calibration procedure of the regulation system of the pumps, the uncertainty on the inflow discharge is the same as for the outflow discharge (3.5 %).

The water depths in the streets were measured by an optical gauge fixed on an automatic traverse system. The resulting measurement uncertainty depends on the instrument accuracy (± 1 mm) and on the fluctuations of the free surface. Throughout most of the experimental model, these fluctuations did not exceed 1 mm (2 mm for the considered highest discharge). The fluctuations reached locally 4 to 10 mm downstream of the intersections between the main streets (4, C and F) and downstream of the intersections located close to the south-east corner of the model (Araud, 2012).

### 2.1.3 Test program

Two main series of experiments were considered here. In the first series, the total inflow discharge was varied from 10 m$^3$/h up to 100 m$^3$/h, with three intermediate values of discharge (20, 60 and 80 m$^3$/h) as detailed in Table 1. In all the tests of this first series, the partition of inflow discharge was kept equal between the west and the north faces. These discharges are consistent with those observed in previous flood events in dense urban floodplains (Mignot et al., 2006), scaled according to the Froude similarity. In the second series of tests, the total inflow was kept at a constant value (60 m$^3$/h) and the inflow partition between the west and north faces was varied systematically from 0 % up to 100 % by steps of 10 % (Table 1).

The outflow discharge was measured downstream of each street for all the tests. In addition, for the tests of the first series of experiments (except for 10 m$^3$/h), the water levels were measured along all the streets.

The reproductibility of the experiments was tested by comparing water depth and discharge measurements on experiments repeated at several day intervals. The upstream and downstream boundary conditions were identical for all the repetitions. For instance, for test Q020-050, a difference of less than 1 mm was found on 90 % of replicate measurements of water depth. This difference corresponds to the resolution of the measuring device. 96 % of the differences were lower than 2 mm, which corresponds roughly to the fluctuations of the free surface for this test. The outflow discharges in all the streets were measured 13 times. The observed differences were also of the order of the measurement uncertainties (Araud, 2012).

In the laboratory experiments, the Reynolds numbers $R = 4\,h\,u\,/\,\nu$ takes values of the order of $10^4 \div 10^5$ ($h$ is the water depth, $u$ the depth-averaged velocity and $\nu$ the kinematic viscosity of water). At the prototype scale, $R$ is expected to reach $10^6 \div 10^7$, as discussed in Sect. 5.1.



## 2.2    Numerical model

The ability of a standard shallow-water model to predict the observed discharge partitions and water depths was tested using the numerical model WOLF 2D. This model has been developed by the research group HECE of the University of Liege (Belgium). It solves the fully dynamic shallow-water equations on multiblock Cartesian grids and a two-length scale $k$-$\varepsilon$
turbulence model is used to account for the anisotropic turbulent mixing induced by the lateral shear and by the bottom-generated turbulence (Camnasio et al., 2014; Erpicum et al., 2009). The numerical discretization is based on a conservative finite volume scheme and a self-developed flux vector splitting (Dewals et al., 2008; Erpicum et al., 2010b).

In 2003, the model WOLF 2D was selected by the regional authorities in Belgium to perform all detailed 2D flow simulations to support official inundation mapping, including in the framework of the European Floods Directive. Since
then, it has been routinely applied for inundation modelling.

In the applications considered here, the bottom shear stress was estimated using Darcy-Weisbach formulation and the friction coefficient was evaluated by Colebrook formula as a function of a roughness height $k$ defined by the modeller. This formulation was preferred here to Manning formula because it is more process-oriented and enables therefore a more objective estimation of the corresponding roughness parameter (roughness height vs. Manning coefficient). Unless otherwise
stated, all simulations were performed with a roughness height of zero, consistently with the smooth walls and bottom of the experimental model.

The geometry of the scale model was implemented in the numerical model by using the building hole method (Schubert & Sanders, 2012). The cell size is uniform and was taken equal to $1 \times 1$ cm². This choice was made to obtain a realistic number of cells over the width of each streets (about 7 to 12) compared to the relative grid resolution used in practice for inundation
modelling. Results obtained with a cell size of $5 \times 5$ mm² are also discussed (Sect. 3.2).

The inflow discharge was prescribed as a boundary condition upstream of each street, while a free flow was considered at the downstream boundaries. The boundary conditions for the turbulence model were set according to Camnasio et al. (2014) and Choi & Garcia (2002).

## 3    Influence of inflow discharge

In a first test series of tests, the total inflow discharge was varied from 10 to 100 m³/h, while preserving an equal distribution of inflow discharge between the west and north faces. In Sect. 3.1, we describe the experimental observations and the computational results based on the numerical model introduced in Sect. 2.2. Next, we show the influence of modelling choices such as the grid refinement, the turbulence model and the roughness parameter (Sect. 3.2).



## 3.1 Experimental observations and numerical results

We present hereafter the experimental and numerical results in terms of discharge partition between the south and east faces, as well as at the street level. We also describe the observed and computed water depths, both at the district and at the street levels. In terms of numerical results, this section discusses only the results obtained by using a cell size of 1 cm, the $k$-$\varepsilon$ turbulence model and Colebrook friction formula with a roughness height equal to zero. Variants of this are detailed in section 3.2.

### 3.1.1 Discharge partition between the south and east faces

The circular markers in Figure 2 represent the observed partition of the outflow discharge between the east and south faces of the urban district for five different total inflow discharges. Overall, about 60% of the outflow discharge crosses the south face and 40% the east face. The outflow partition remains virtually independent of the total inflow discharge. This is a first distinctive result of the experiments. For a total inflow discharge varying by one order of magnitude, from 10 m³/h up to 100 m³/h, the portion of outflow through the east face varies only from 38.6% to 40.6%.

The shallow-water model succeeds in predicting the overall discharge partition between the south and east faces (Figure 2). The difference between the observed and computed values remains in the range 0.02-2.16 %. So, the numerical model also reproduces the quasi independence of the outflow discharge partition with respect to the total inflow.

### 3.1.2 Discharge partition at the street level

Figure 3 details the partition street by street of the outflow discharge for five different inflow discharges. The highest outflow discharges correspond to the widest streets, particularly those which are straight and start more upstream in the model i.e. closer to the north-west corner (streets C and 4).

At the outlet of street C, which is 2.5 times wider than streets A and B, the observed outflow discharge is about 2.3 to 2.5 times higher than the corresponding discharges in streets A and B. This leads to similar unit discharges in the different streets and may results from the similar configurations of streets A, B and C in terms of shape (all three are straight) and encountered types of intersections. In contrast, street 4 collects in-between 2.0 and 3.0 times more discharge than streets 1, 2 and 3, while the ratio of the street widths is also 2.5. These larger deviations may result from the different configurations of streets 1, 2 and 3 compared to street 4, since the latter is straight while the former are curved, leading to different types of intersections.

Another example of influence of the shape of the streets and intersections may be noticed by comparing streets 1 and A. Their inflow discharges are the same; but street A has an observed outflow discharge in-between 60 % and 70 % higher than the observed outflow in street 1. The number of intersections is the same for both streets. However, street A has mostly right-angle intersections while all intersections in street 1 have different angles, which seems to promote more flow to be diverted towards the lateral streets. The difference in the outflow discharges results most likely from this difference in the shapes of




the streets. Similarly, street F, which is as wide as street C, discharges at the outflow only about 55-58 % of the discharge from street C, as street F is curved and located further from the "upstream" corner (north-west).

Similarly as for the discharge partition between the faces, the observed portion of outflow discharge in each street remains essentially independent of the total inflow (Figure 3). For a total inflow varying by one order of magnitude (from 10 m³/h to 100 m³/h), the scaled sensitivity of the outflow discharges in the different streets is in average 4% and it does not exceed 12%, except in street G where it reaches 19 %.

In the computed results, the outflow from the streets with the highest discharges (4 and C) are overestimated by 10-30 % compared to the experimental results. The opposite is observed for some of the streets with the lowest discharges (1-3, D-F), while the outflows from streets 5-7 and G are fairly well represented by the numerical model. The outflow discharges from the streets with intermediate discharges (A, B and F) are also generally well predicted by the model.

As the obtained discrepancies are maximum in curved streets (1, 2 and 3), it is likely that they partly result from the Cartesian grid used, which relies on a "staircase" approximation of the obstacles not aligned with the grid. A Cartesian grid remains however of high relevance in practice (Kim et al., 2014), as it makes it generally straightforward to handle contemporary gridded data obtained from remote sensing technologies (e.g., Light Detection And Ranging, LIDAR).

Another possible explanation for the discrepancies stems from the complexity of the actual flow fields at the intersections, involving different flow regimes, hydraulic jumps and waves as described in the literature cited in the Introduction section. Here, it is however difficult to identify which intersection is responsible for the main discrepancies as they all interact with each other and the experimental flow partition between the streets is only available at the downstream end of each street and not in-between all the intersections.

A closer look at the computed results reveals that the higher the total discharge, the higher the outflow from the widest straight streets 4 and C (+ 0.6 % and + 1.75 % respectively for a total inflow varying from 10 m³/h to 100 m³/h). In the streets with the lowest outflow discharges (1-3, 5-7, D, E and G), the computed variation is opposite: minus 0.4 % in average. In contrast, the variations observed experimentally are not monotonous as the numerical results are (e.g., in streets 1, 3, 4 and C).

### 3.1.3 Water depths at the district level

For four different inflow discharges, maps of the observed and computed water depth distributions over the whole district are provided as Supplement 1, both in absolute values and scaled by the district-averaged water depth (Figs. S1 and S2). The observed district-averaged water depth is shown to increase from 3.4 cm up to 9.0 cm when the inflow discharge is varied from 20 m³/h to 100 m³/h (Table 2). The relative distribution of water depths across the district is hardly affected by the total inflow, as shown in Supplement 1 (Fig. S1).

The computed values are in excellent agreement with the observations for the lowest inflow discharge; but they deviate by about 11 % for the highest inflow discharge (100 m³/h, Table 2). For this discharge, the observed maximum water depth is



14.1 cm, while the corresponding computed value reaches 15.72 cm. This also hints that the Cartesian grid may explain part of the discrepancies, as this mesh effect is expected to lead to greater overestimations as the flow velocity increases.

### 3.1.4    Water depths at the street level

The profiles of observed water depths are displayed in Figure 4 for the two widest streets (4 and C). The most significant

variations in the water depths take place locally, in the vicinity and immediately downstream of the intersections, particularly close to the intersections of two wide streets such as streets C and 4, or streets 4 and F. In contrast, in-between the intersections, the water depths remain fairly constant. Therefore, friction is expected to play a minor part, as discussed in Sect. 3.2.

As shown in Figure 5 and Figure 6, the computed results reproduce qualitatively the main features of the water depth

profiles, which are characterized by sudden drops near the intersections and remain almost constant in-between the intersections. From a quantitative perspective, the computed results are relatively accurate for the lowest discharge (20 m³/h); but they tend to deviate from the observations as the inflow discharge is increased. This is also highlighted in Figure 7(a) and (b), which shows an increase in the root mean square (RMS) error from about 1-2 mm up to 1-1.4 cm as the inflow discharge rises from 20 m³/h up to 100 m³/h. As shown by the sign of the bias in Figure 7(c) and (d), all significant deviations

correspond to overestimations of the water depths. Again, this trend is consistent with an extra flow resistance induced by the Cartesian grid when the obstacles are not aligned with the grid.

### 3.2    Sensitivity analysis of the numerical results

To appreciate the sensitivity of the results to different modelling choices, a sensitivity analysis was performed to investigate the influence of roughness, mesh refinement and the turbulence model.

### 3.2.1    Roughness parameter

The reference simulations were carried out with a roughness height equal to zero ($k = 0$ mm) as the experimental model was in Plexiglas, which is similar to glass ($k \sim 0.01$ mm) in terms of roughness characteristics. The simulations were also repeated with a roughness height $k = 1$mm, which we considered as an upper bound of the range of possible roughness characteristics of the experimental setup. Testing higher values of $k$ was deemed unrealistic.

The distribution of the outflow discharges between the different streets remains virtually unchanged when the roughness height is varied between 0 and 1 mm (Fig. S3 in Supplement 2). For $k = 0$ mm, the RMS error of the computed outflow discharges ranges between 1.7 % and 2.0 % of the inflow discharge, while it ranges between 1.7 % and 1.9 % when the roughness height is increased up to 1 mm (Table S1 in Supplement 2). This confirms the slight influence of the roughness parameter on the computational results, as anticipated from the water profiles measured in the main streets of the

experimental model (Figure 4). Whether this finding also applies at the prototype scale is discussed in Sect. 5.



### 3.2.2 Grid refinement

The influence of the mesh refinement on the results was tested by using cell sizes of 1 cm, 5 mm and 2.5 mm. For total inflows of 10, 20, 60, 80 and 100 m³/h, Figure 8 compares the computed and measured outflow discharges in each street. Most points fall within the +/- 15 % range in terms of relative error street by street. For a grid spacing of 1 cm, the root mean square error on the outflow discharges lies between 1.7 % and 2.0 % of the total inflow (Table 3). Shifting from a 1 cm grid to a 5 mm grid leads to a reduction in the root mean square error on the discharges to 1.1 % - 1.5 %. Figure 8 also shows that the discharge partitions in the widest streets are substantially improved when the cell size is reduced to 5 mm.

For a total inflow of 20 m³/h, the computation was also performed with a 2.5 mm grid; but the change in the root mean square is slight (below 0.1 %). Thus, grid refinement to 2.5 mm does not lead to a significantly better fit of the model predictions. This suggests that the numerical discretization with a cell size of 5 mm is "converged" in the sense of a grid convergence analysis (Roache 1994).

The sensitivity of the computed water depths with respect to the grid resolution was also assessed. As shown in Figure 7, refining the cell size to 5 mm instead of 1 cm reduces the RMS error and the bias by about 20 % in the case of the highest inflow discharges, for which these errors are maximum. This reduction is in agreement with a decreased influence of the staircase approximation of the obstacles geometry when a finer grid resolution is used.

Changing the grid size leads also to local changes in the flow pattern. As an example, Figure 9 shows the details of the flow field near the downstream end of street 4 in the case of test Q100-W050 computed with cell sizes of 1 cm and 5 mm. The discretization of the model geometry on a Cartesian grid induces local discontinuities in the street widths as long as these streets are not perfectly aligned with the grid, as it is the case here particularly because the "as-built" coordinates of the obstacles were used. These sudden changes in the street width lead in some cases to the development of flow structures (such as cross waves), which as a matter of fact are mesh-dependent (Figure 9a and b). Their impact remains however very limited further upstream in the domain, where the flow patterns are extremely similar for the two grid sizes (Figure 9).

Despite the better results obtained with the cell size of 5 mm, in most of the simulations performed hereafter, the 1 cm grid was kept nonetheless because in our opinion this choice is the most consistent with grid refinements reasonably accessible for inundation mapping in practice. It corresponds indeed to five cells over the width of the narrow streets and about twelve cells over the width of the wide streets (4, C and F, as detailed in Table 4). At the field scale, it leads to a grid spacing of 2 m (Sect. 5.1). Using finer cells (5 mm or 2.5 mm) would not be realistic compared to typical grid refinements used for real-world flood hazard mapping. The 1 cm grid may also be considered as a reasonable trade-off between accuracy and computational burden since opting for the 5 mm grid would increase the computational cost by almost one order of magnitude (~ 8) for a moderate reduction in the errors.





### 3.2.3  Turbulence model

Most flood hazard mapping in practice is conducted with depth-averaged models which do not incorporate a proper turbulence model, apart from a friction term which lumps all dissipative effects. Here, we specifically tested the influence of activating or not the $k$-$\varepsilon$ turbulence model which is available in WOLF 2D.

The computations reveal that the turbulence model hardly influences the outflow discharges. This result applies when the outflows are examined by face (Figure 2) and also when they are disaggregated at the street level (Figure 8). At the face level, switching on or off the turbulence model induces variations in the outflow discharge not exceeding 1.5 % (Figure 2). At the street level, the relative differences may reach 2 %; but these differences are much lower than those observed when changing the grid size from 1 cm to 5 mm. For the whole range of considered total inflows, the influence of the turbulence

model on the root mean square error of the outflow partition between the streets is of the order of 0.1 % of the total inflow, both for the 1 cm grid and for the 5 mm grid (Table 3 and Table S1 in Supplement 2). This remains about five times lower than the influence of the grid size, but similar to the influence of the roughness parameter $k$.

As regards the influence of the turbulence model on the computed water depths, the simulations without turbulence model perform slightly worse than when the turbulence model is used (Figure 5 and Figure 6). This is also confirmed by an increase

in the RMS error and in the bias for most considered discharges (Figure 7).

Like the cell size, the turbulence model has also a substantial influence on some local features of the flow field. For tests Q020-W050 and Q100-W050, Figure 10 shows that the recirculation lengths in the streets differ significantly between the results computed with and without the turbulence model. In the former case, the recirculation length $R_T$ is of the order of two to four times the street width (Figure 10a, c) and is in overall agreement with results of detailed 3D simulations performed by

Li & Zeng (2010), also in terms of recirculation width. In contrast, simulations without the turbulence model lead to recirculation zones which tend to preserve a constant width over the whole length of the streets (Figure 10b, d) and their overall shape looks less realistic, particularly in comparison with the results of Li & Zeng (2010). This results from the hardly dissipative nature of the flow model without turbulence model.

Such changes in the flow structure between simulations with and without turbulence may be related to local variations in the

discharge partition. For test Q100-W050, Figure 11 details the flow field at the intersection between streets 4 and F. Besides variations in the recirculation lengths as already highlighted in Figure 10, Figure 11 also reveals a change in the recirculation width in the southern branch. While the width reaches 6-7 cm when the turbulence model is used (Figure 11a, c), it is restricted to 3-4 cm without turbulence model. As a result, the share of discharge maintained in street 4 in the simulation with turbulence is about 9 % higher than in the simulation without turbulence. This figure was obtained with a grid size of

5 mm, while the change is estimated at 6 % based on the 1 cm grid size. In turn, this change in the flow partition alters the shape of the control sections in the downstream part of street 4 (Figure 11a, b), and also the overall distribution of water depths.



In the following, the turbulence model has been systematically used due to the higher realism that it provides for simulating the flow processes.

## 4    Influence of inflow partition

A distinctive feature of the present experimental setup is that it enables the partition of inflow to be varied continuously between the west and the north faces. For a total inflow discharge of 60 m³/h, the portion of inflow through the west face (noted $\phi_{west}$) was varied between 0 % and 100 % by steps of 10 % (Table 1). In these tests, only the outflow discharges were measured and not the water depths.

As shown in Figure 12, the observed outflow discharge through the east face (noted $\phi_{east}$) decreases as $\phi_{west}$ is reduced. However, for $\phi_{west}$ varying from 100 % down to 0 %, the observed value of $\phi_{east}$ is only reduced from 45 % down to 37 %. So, the scaled sensitivity of $\phi_{east}$ with respect to $\phi_{west}$ is lower than 0.1, reflecting the ability of the multiple obstacles faced by the flow within the district to redistribute the outflow discharge at the face level almost irrespective of the upstream flow partition.

The computed outflow discharges through the east face slightly underestimate the observed values (by 1.5 % to 3.3 %) but they show a similar trend as the observations: $\phi_{east}$ drops by 10 % (from 43 % down to 33 %) when $\phi_{west}$ is reduced from 100 % to 0 %.

The distribution of the outflow discharges at the street level is shown in Figure 13. Three groups of streets may be distinguished:

- in streets 1 to 4, which end up on the eastern face, the outflow discharge declines steadily as $\phi_{west}$ is increased;
- contrarily, the outflow discharges rise in street C and to a lesser extent in streets A, B, D and F, which all end up on the southern face;
- the other streets are less significantly influenced by the partition of outflow discharge.

Figure 14 compares the computed results to the observations. Consistently with the results shown in Figure 3, the computations overestimate the outflow discharges in the widest streets (by about 20 % in streets C and 4), while they underestimate the outflow discharges in most other streets. This aspect is shown here to remain similar independently of the partition of the inflow discharge (Figure 14). Despite the overestimation of the outflow discharge in streets C and 4, the relative influence of a change in $\phi_{west}$ is fairly well reproduced by the numerical model.



## 5    Discussion

### 5.1    Upscaling

The simulations presented in Sect. 3.2 have revealed that the value of the roughness parameter $k_s$ and the turbulence model have little influence on the computed discharge partitions and water depths in the laboratory model. We discuss here to which extent these conclusions remain valid for real-world applications, i.e. at prototype scale. Assuming Froude similarity, the simulations were repeated at the prototype scale following three different procedures, referred to as Prototype 1 to 3 in Table 5.

First, all geometric characteristics of the laboratory model were magnified by a factor $e_H = e_V = 200$, leading to street widths of the order of 10 to 25 m (Prototype 1 in Table 5). Using Colebrook friction formula (or the Moody diagram), the roughness height $k_s$ was selected so as to preserve approximately the same Darcy-Weisbach coefficient ($f \sim 0.02$) as at the scale of the laboratory model. Note that $k_s$ does not simply scale with the water depth since the laboratory model operates in the transition zone (R $\sim 10^4 \div 10^5$), while the flow is fully turbulent at the prototype scale (R $\sim 10^7 \div 10^8$). In the end, the simulations results obtained at the prototype scale were divided by the factor $e_H = e_V = 200$ to enable direct comparisons with the results at the scale of the laboratory model. As shown in Fig. S4 (Supplement 3), the two sets of results match almost perfectly. This agreement was obtained for roughness heights at the prototype scale between zero and up to $k_s = 5$ cm, with and without turbulence model, confirming that in Prototype 1 the value of $k_s$ and the turbulence model have little influence on the results, consistently with the findings at the laboratory scale.

However, Prototype 1 is hardly realistic in terms of height to width ratio in the streets. Most real-world urban floods are characterized by much smaller water depths compared to the widths of the streets. Therefore, upscaling of the laboratory model was also performed assuming a distorted model with a horizontal scale factor $e_H = 200$ and a vertical scale factor $e_V = 20$ (Prototype 2 in Table 5). In Prototype 2, a roughness height of the order $5 \times 10^{-3}$ m leads to similar values of the Darcy-Weisbach coefficient as at the laboratory scale. Nonetheless, as a result of the smaller water depths, the relative effect of friction becomes higher than in the experimental configuration, as shown in Fig. S4. Finally, we consider that a reasonable roughness height characterizing urbanized floodplains is of the order of $\sim 0.1$ m (Prototype 3 in Table 5) instead of $5 \times 10^{-3}$ m. As shown in Fig. S4, this further emphasizes the influence of friction and dissipation on the flow, which is not properly reflected in the experimental model made of smooth boundaries (Plexiglass).

### 5.2    Porosity-based approach

In Sect. 3.1, the discrepancies between computed and observed results were to a large extent attributed to the intrinsic limitations of Cartesian grids to reproduce oblique boundaries. To investigate this effect further, we tested an extended shallow-water model involving porosity parameters to improve the representation of complex boundaries in a Cartesian grid framework. This approach is similar to the *cut-cell* technique (An et al., 2015; Causon et al., 2000; Kim & Cho, 2011).



We used two types of porosity parameters (Figure 15). First, a *storage porosity* $\phi$ is defined for each cell $(i,j)$ of the two-dimensional grid. It represents the fraction of the cell surface which is actually available to store water despite the presence of solid obstacles (i.e. the void fraction in the cell). Similarly, a *conveyance porosity* $\psi$ is defined at each cell interface to reflect the blockage effect of the solid obstacles. It quantifies the fraction of the face length which remains available for

exchanges with the neighbouring cells. The resulting shallow-water equations with porosity are formally equivalent to those used by Sanders et al. (2008) and Kim et al. (2015).

The partition of outflow discharges at the street level has been computed using the model with porosity. The results are displayed in Figure 16, which should be compared to Figure 3. While the outflow from streets 4 and C are overestimated by 10-30 % when the standard shallow-water model is used, this discrepancy is reduced here to around 10 %. Similarly, the

10 outflow discharges through the narrower streets 1-3 and D-F were significantly underestimated by the standard model, whereas these outflows are now predicted with an error not exceeding 8 %. This leads to a root mean square error on the outflow discharge which is reduced from 19 % down to 6.6 % as a result of using the shallow-water equations with porosity.

## 6    Conclusion

To investigate the flow characteristics in urbanized floodplains, we considered here the 5 m × 5 m Icube Laboratory model

of a typical European urban district. The model involves 14 streets of different widths (5 cm or 12.5 cm). By means of a system of 14 regulated pumps, the inflow discharges could be controlled individually within each street of the district. This was not possible on earlier experimental setups reported in literature. The outflow discharges were measured downstream of each street using calibrated weirs and the water profiles in the streets were measured using an optical gauge mounted on an automatic traverse system. The experimental observations were compared with the results of an existing shallow-water

model which was used earlier for real-world inundation mapping. The numerical model is based on a Cartesian grid.

Two test series were considered. In the first one, 50 % of the inflow discharge was injected through the west face of the model and 50 % through the north face. The partition of the outflow discharge between the downstream faces (east and south) did not change by more than 2 % as the total inflow was varied by one order of magnitude (between 10 m³/h and 100 m³/h). Similarly, the observed portion of outflow discharge in each street remains virtually independent of the total

25 inflow. The numerical model succeeds in predicting accurately the flow partition between the east and south faces, whereas it overestimates the outflow through the widest streets (by 10 to 30 %) and underestimates the outflow through narrower streets.

Drops in the free surface profiles were observed downstream of each intersection. In-between the intersections, the water depths remain fairly constant. In general, the computed water depths overestimate the experimental observations, by 1 %

(lowest total inflow) up to about 10 % in average (highest total inflow). This overestimation of the overall flow resistance is consistent with the staircase representation of complex geometries when using a Cartesian grid.





As shown by a sensitivity analysis, the roughness parameter has very little influence on the computed results. The same applies for the turbulence model, except for some features of the velocity field which are predicted more realistically when a depth-averaged $k$-$\varepsilon$ turbulence model is used (e.g., shape and length of recirculation zones). Some influence of the cell size remains; but the cell size was chosen purposely to correspond to a grid refinement which may realistically be used in practice

(about 5 to 12 cells over the width of the streets).

In a second series of tests, the partition of the inflow discharge between the west and north faces was varied systematically by steps of 10 % between $\phi_{west} = 0$ % (inflow through the north face only) and $\phi_{west} = 100$ % (inflow through the west face only) for a constant total inflow. The relative influence of a change in $\phi_{west}$ was fairly well reproduced by the numerical model, despite remaining overestimations of the outflow discharge through the widest streets and vice-versa for the narrower

streets.

The upscaling of the present findings to real-world applications was discussed. In particular, the influence of the roughness parameter is expected to become significantly stronger at the prototype scale due to the combined effect of a lower height-to-width ratio of the wetted sections and a relatively higher roughness height characterizing real urban settings.

The substantial influence of the Cartesian grid on the flow computations due to a staircase representation of the obstacles

geometry was confirmed by implementing a subgrid model based on porosity parameters. This enhanced shallow-water model lead to substantially better predictions of the outflow discharges at the street level. However, this approach is not used routinely for operational inundation mapping.

In the future, the present research will be extended to investigate the hydrodynamic characteristics of unsteady inundation flow in urbanized floodplains, as well as the influence of the bottom slope. Several other processes have not yet been

considered and should be addressed in subsequent studies, such as flow within the buildings, morphodynamic changes and the transport of debris by the inundation flow.

**Acknowledgement**

The Icube laboratory experimental model was funded by the Alsacian network of laboratories in Environmental Engineering and Sciences and the Fluid Mechanics research team of the Icube laboratory. The authors are grateful to the Icube laboratory

technicians (Martin Fisher, Johary Rasamimanana and Abdel Azizi) and the PhD students (Hakim Ben-Slimane, Vivien Schmitt, Noëlle Duclos, Alain Petit-Jean and Sandra Isel) for their priceless contribution to the building of the experimental setup. A special thank is also addressed to Quentin Araud who conducted most of the experiments.

Part of this research was funded through the ARC grant for Concerted Research Actions, financed by the Wallonia-Brussels Federation. The Authors gratefully acknowledge Mathieu Debaucheron who contributed to some of the numerical

computations.





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



**Tables**

Table 1 : Test series dedicated to testing the influence of the total inflow of the inflow partition between the west and north faces.

| | Test ID | Total inflow discharge (m³/h) | Portion of inflow through the north face (%) |
|---|---|---|---|
| Test series n°1 | Q010-W050 | 10 | 50 |
| | Q020-W050 | 20 | 50 |
| | Q060-W050 | 60 | 50 |
| | Q080-W050 | 80 | 50 |
| | Q100-W050 | 100 | 50 |
| Test series n°2 | Q060-W000 | 60 | 0 |
| | Q060-W010 | 60 | 010 |
| | Q060-W020 | 60 | 020 |
| | Q060-W030 | 60 | 030 |
| | Q060-W040 | 60 | 040 |
| | Q060-W050 | 60 | 050 |
| | Q060-W060 | 60 | 060 |
| | Q060-W070 | 60 | 070 |
| | Q060-W080 | 60 | 080 |
| | Q060-W090 | 60 | 090 |
| | Q060-W100 | 60 | 100 |

Table 2 : District-averaged water depths for inflow discharges between 20 m³/h and 100 m³/h.

| Test ID | Observed (cm) | Computed (cm) | Relative difference |
|---|---|---|---|
| Q020-W050 | 3.37 | 3.41 | + 1 % |
| Q060-W050 | 6.78 | 7.04 | + 4 % |
| Q080-W050 | 8.25 | 8.55 | + 4 % |
| Q100-W050 | 8.98 | 9.94 | + 11 % |



Table 3: Root mean square error on the outflow discharges in each street (expressed in percentage of the total inflow), as a function of the total inflow and the modelling characteristics.

| Grid spacing | Turbulence model | 10 m³/h | 20 m³/h | 60 m³/h | 80 m³/h | 100 m³/h |
|---|---|---|---|---|---|---|
| 1 cm | None | 2.0 % | 1.8 % | 1.9 % | 1.9 % | 2.0 % |
| 1 cm | $k$-$\varepsilon$ | 1.9 % | 1.7 % | 1.8 % | 1.9 % | 2.0 % |
| 5 mm | None | 1.4 % | 1.2 % | 1.4 % | 1.4 % | 1.5 % |
| 5 mm | $k$-$\varepsilon$ | 1.3 % | 1.1 % | 1.3 % | 1.4 % | 1.5 % |
| 2.5 mm | $k$-$\varepsilon$ | | 1.2 % | | | |

Table 4: Considered grid sizes and corresponding resolution obtained at the field scale.

| Grid spacing (laboratory scale) | Grid spacing upscaled at the field scale | Number of cells over the width of narrow streets | Number of cells over the width of wide streets |
|---|---|---|---|
| 1 cm | 2 m | 5 | 12.5 |
| 5 mm | 1 m | 10 | 25 |
| 2.5 mm | 0.5 m | 20 | 50 |

Table 5 : Characteristic Reynolds number R, roughness height $k_s$ and Darcy-Wesibach coefficient $f$ at the laboratory model scale and at the prototype scale (real-world) as a function of the horizontal and vertical magnification factors $e_H$ and $e_V$.

| | Laboratory model | Prototype 1 | Prototype 2 | Prototype 3 |
|---|---|---|---|---|
| $e_H$ | - | 200 | 200 | 200 |
| $e_V$ | - | 200 | 20 | 20 |
| $k_s$ | $< 10^{-5}$ m | $\sim 5 \times 10^{-2}$ m | $\sim 5 \times 10^{-3}$ m | $\sim 0.1$ m |
| R | $2 \times 10^4 \div 3 \times 10^5$ | $6 \times 10^7 \div 7 \times 10^8$ | $2 \times 10^6 \div 2 \times 10^7$ | $2 \times 10^6 \div 2 \times 10^7$ |
| $k_s / 4\,h$ | $\sim 10^{-5}$ | $\sim 10^{-3}$ | $\sim 10^{-3}$ | $\sim 10^{-2}$ |
| $f$ | $\sim 2 \times 10^{-2}$ | $\sim 2 \times 10^{-2}$ | $\sim 2 \times 10^{-2}$ | $\sim 4 \times 10^{-2}$ |



**Figures**

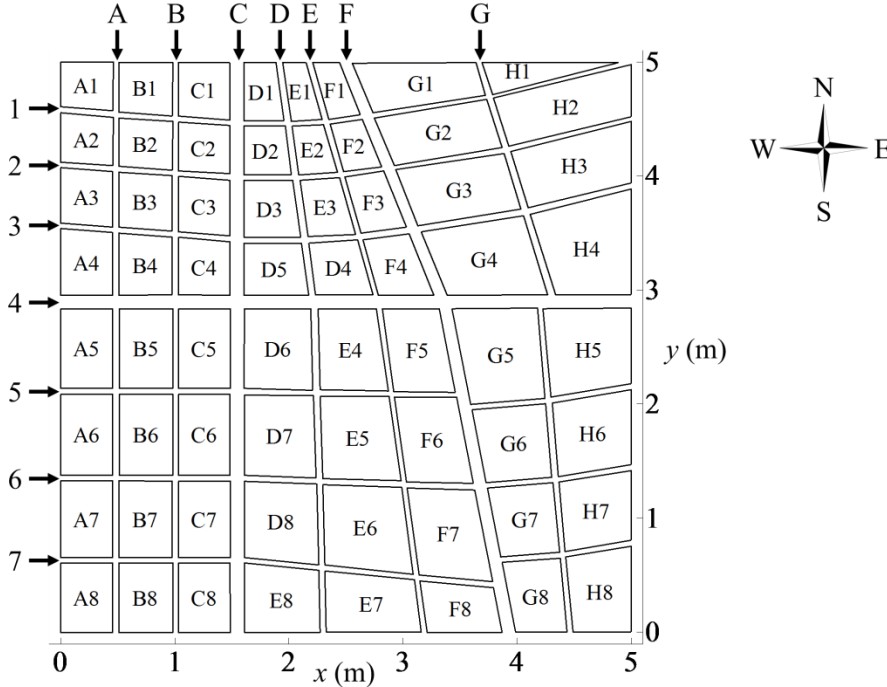

Figure 1: Plane view of the idealized urban district considered in the experiments. Adapted from Araud (2012).

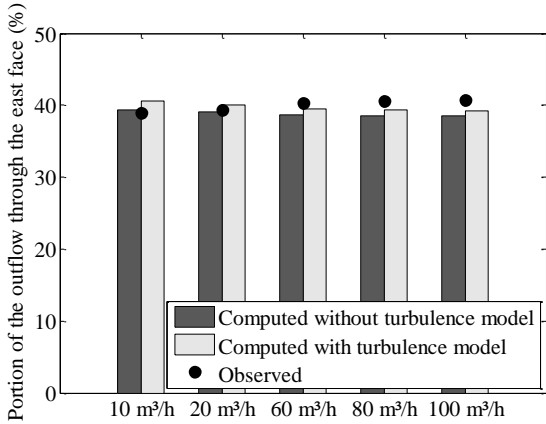

5    Figure 2: Distribution of outflows between the downstream faces in the observations and for different model characteristics.




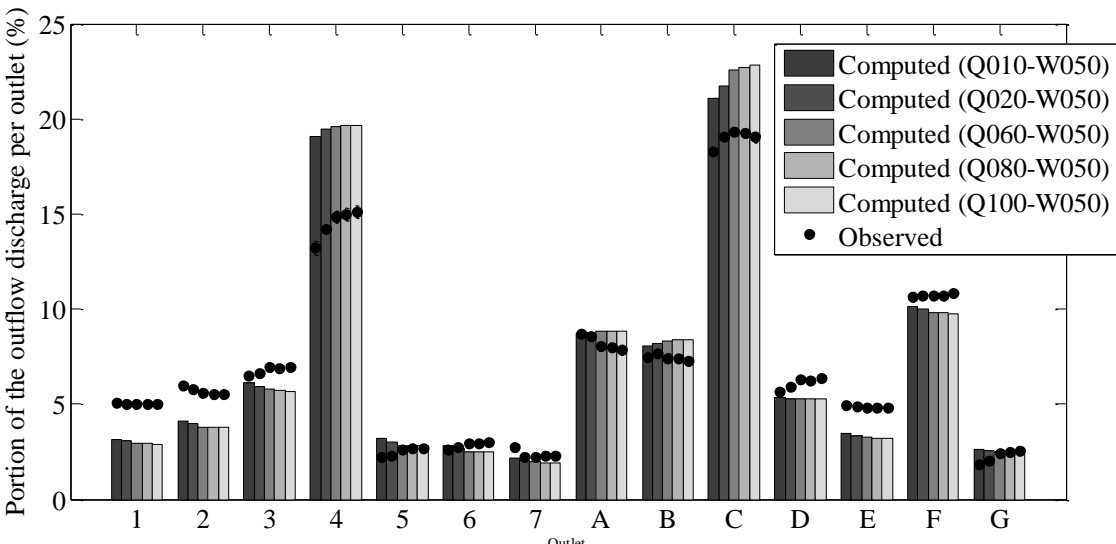

Figure 3: Observed and computed contributions of each street to the total outflow discharge for five different inflow discharges.

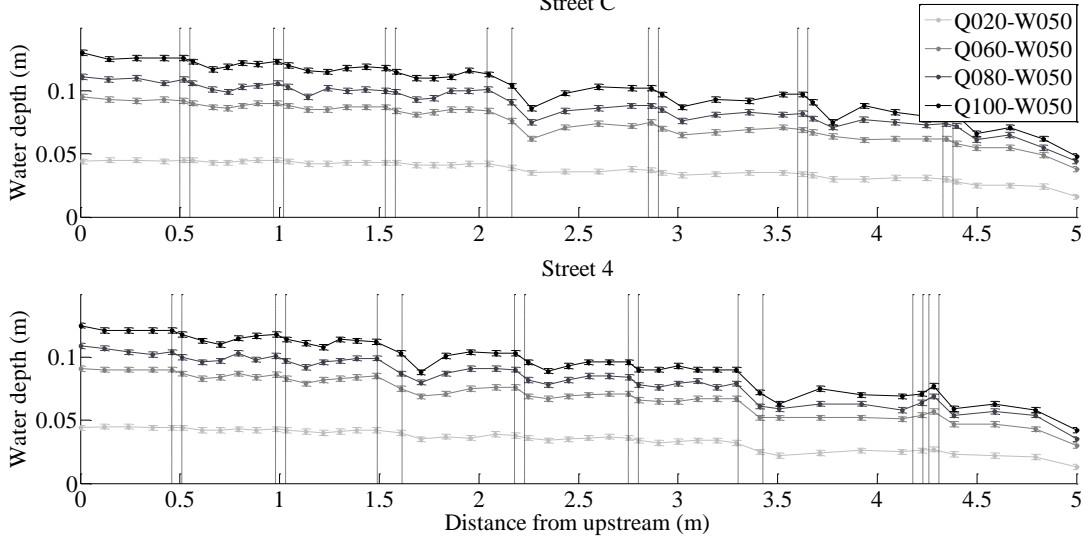

5  Figure 4: Observed water depths in streets C and 4 for inflow discharges varying between 20 m³/s and 100 m³/s. Dashed vertical lines indicate the intersections.




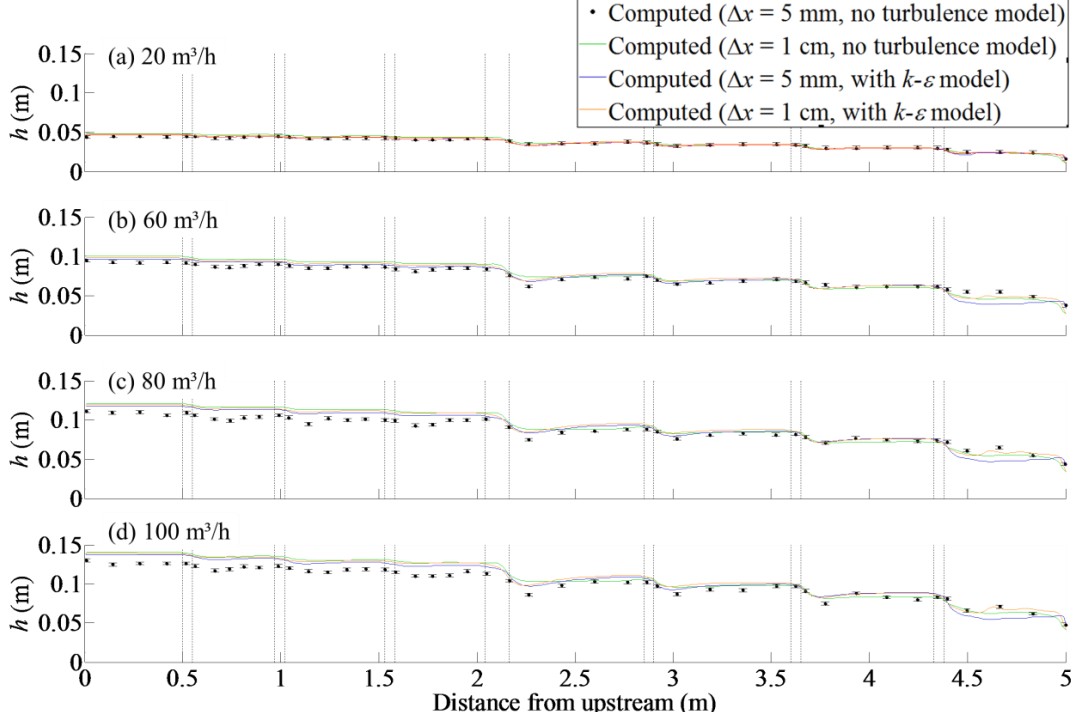

Figure 5: Observed and computed water depths $h$ for inflow discharges varying between 20 m³/h and 100 m³/h in street C.




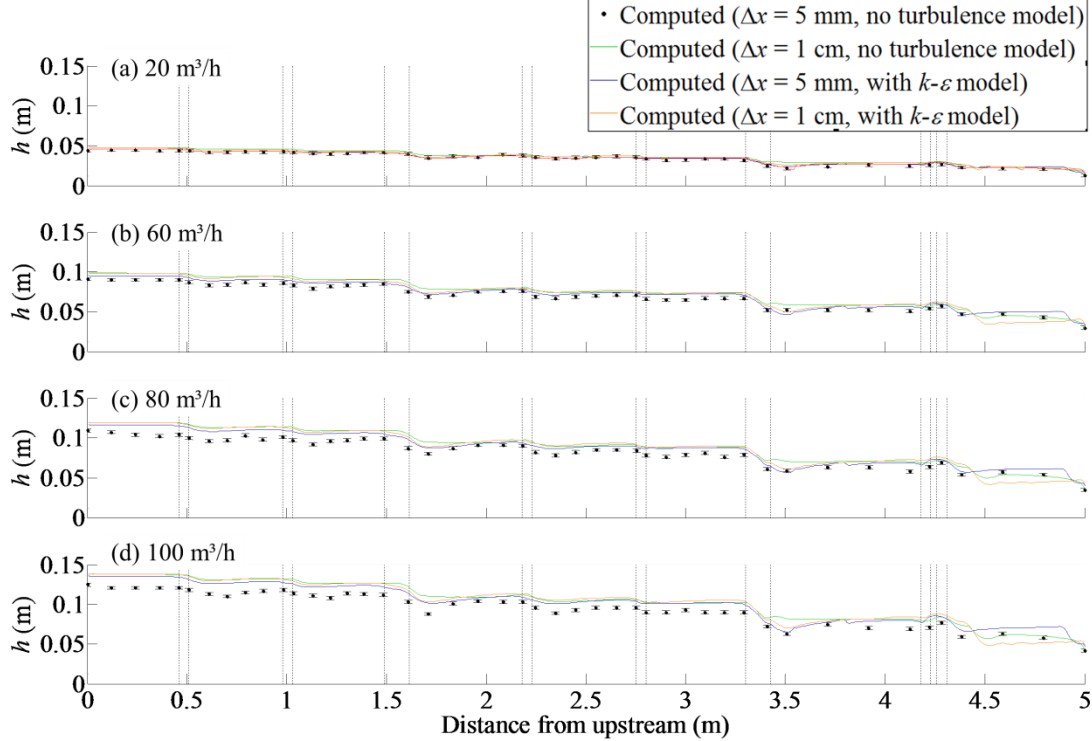

Figure 6: Observed and computed water depths $h$ for inflow discharges varying between 20 m³/h and 100 m³/h in street 4.

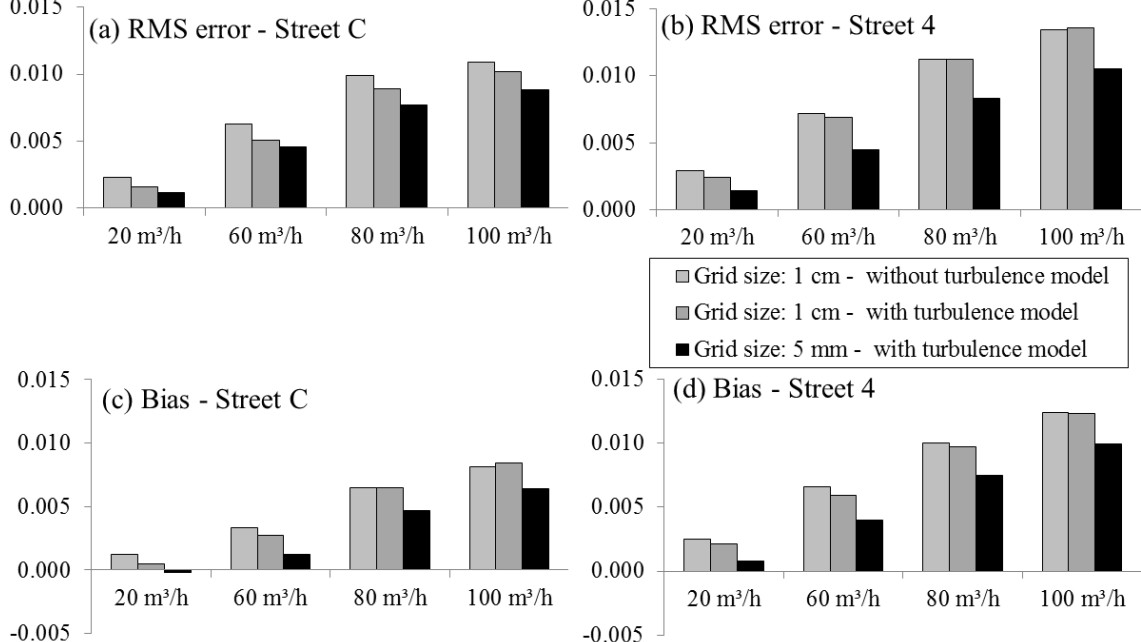

Figure 7: Bias and root mean square (RMS) error on the computed water depths in streets C and 4.

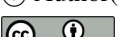



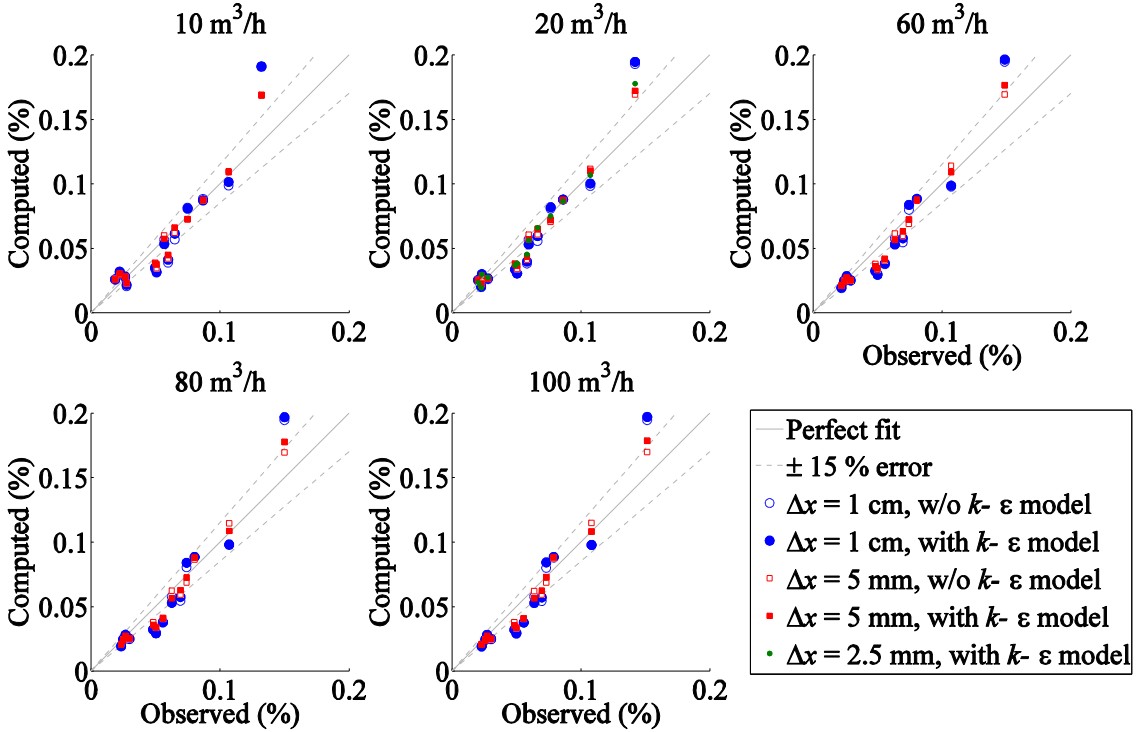

Figure 8: Computed vs. measured outflow discharges in each street (in percent of the total inflow) considering in the computation different cell sizes Δx, with or without turbulence model.

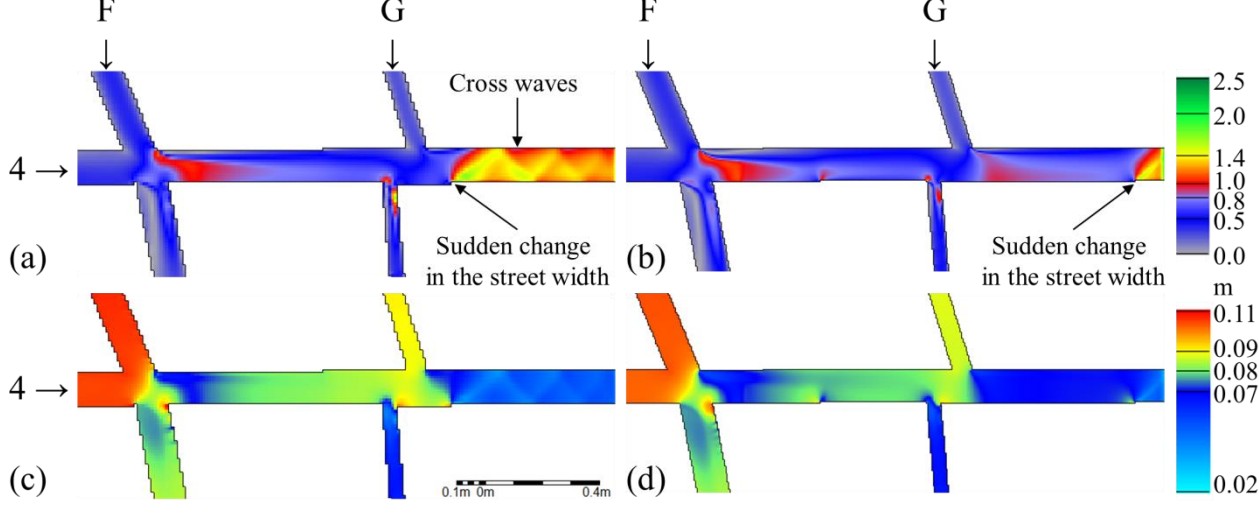

5  Figure 9: Froude numbers (a, b) and water depths (c, d) computed for test Q100_W050 and cell sizes of 1 cm (a, c) and 5 mm (b, d).





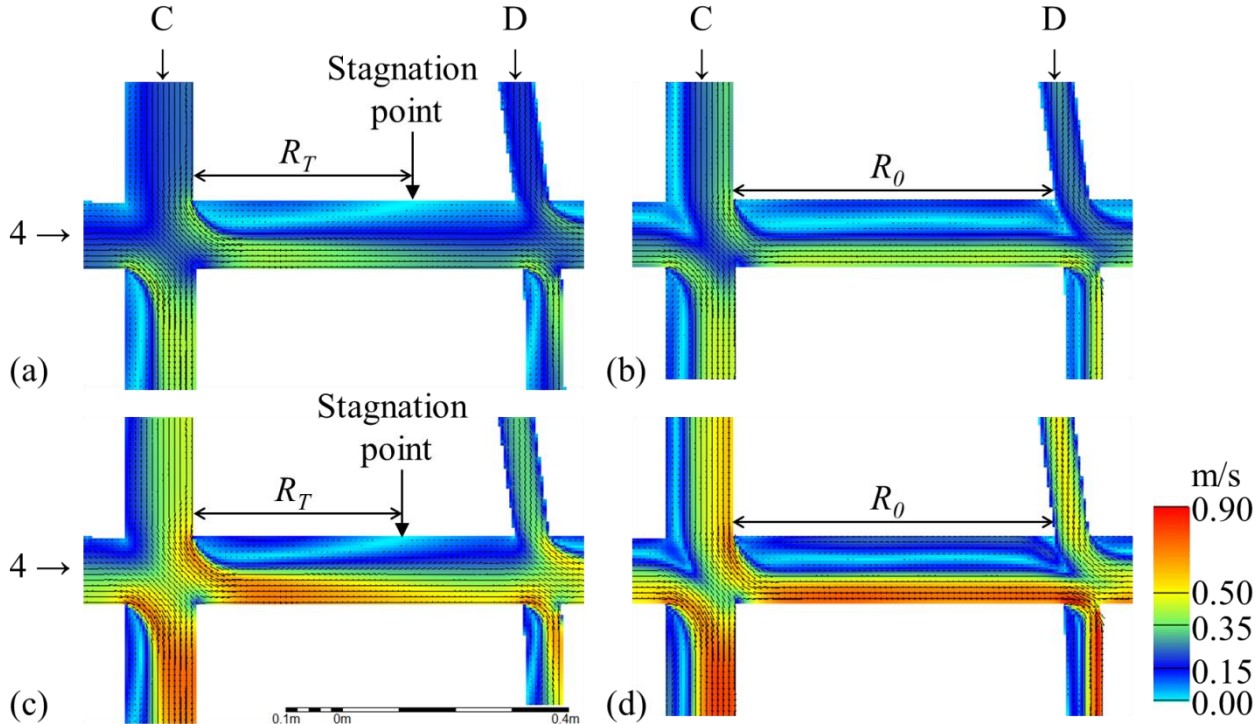

Figure 10: Velocity fields computed with (a, c) and without (b, d) turbulence model for tests Q020-W050 (a, b) and Q100-W050 (c, d). Grid size: 5 mm.





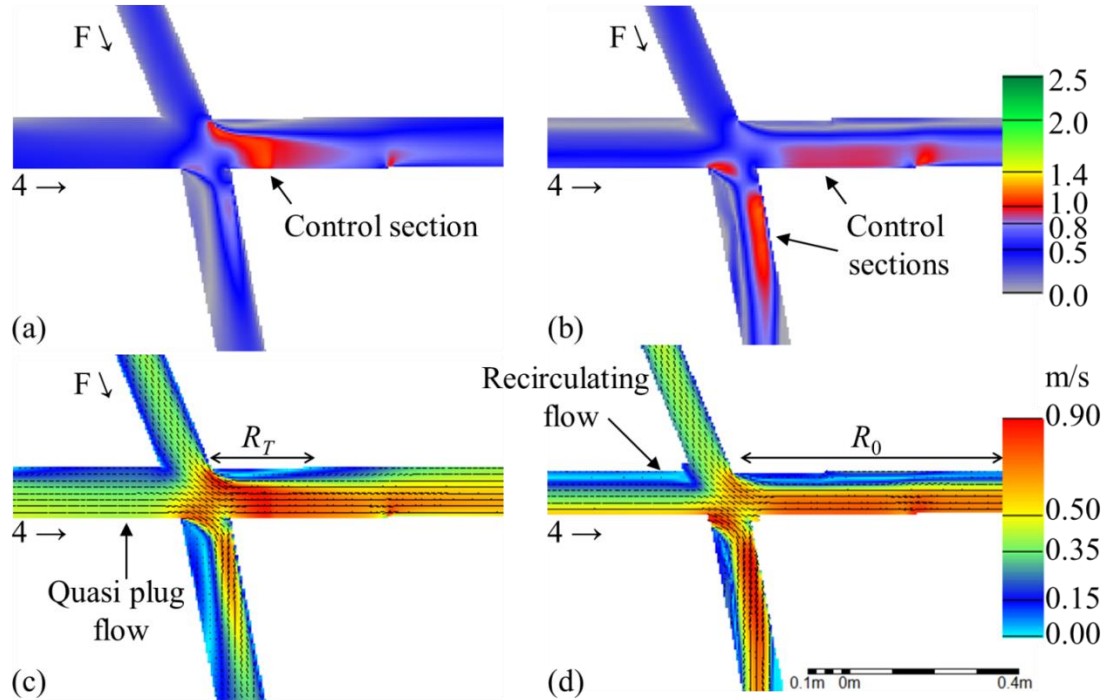

Figure 11: Froude numbers (a, b) and velocity fields (c, d) computed with (a, c) and without (b, d) turbulence model for test Q100-W050 (c, d). Grid size: 5 mm.

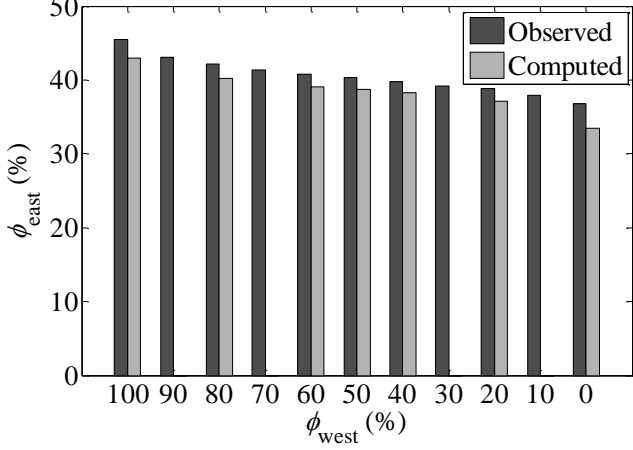

5 Figure 12: Observed and computed portions of outflow discharge through the east face($\phi_{east}$) when the inflow through the north face ($\phi_{north}$) is varied from 0 % up to 100 %.




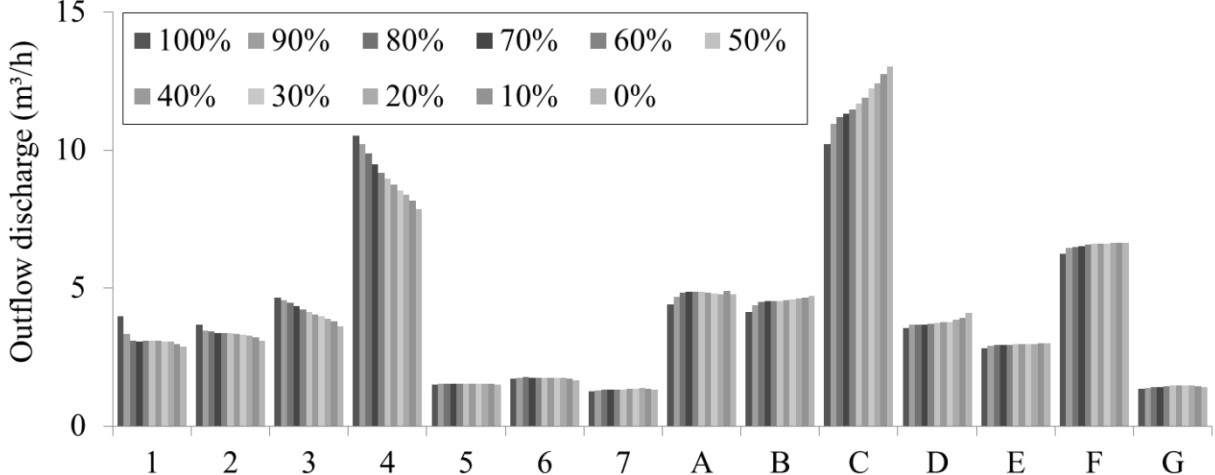

Figure 13: Observed outflow discharges in each street for when the inflow through the west face ($\phi_{west}$) is varied between 100 % and 0 %.

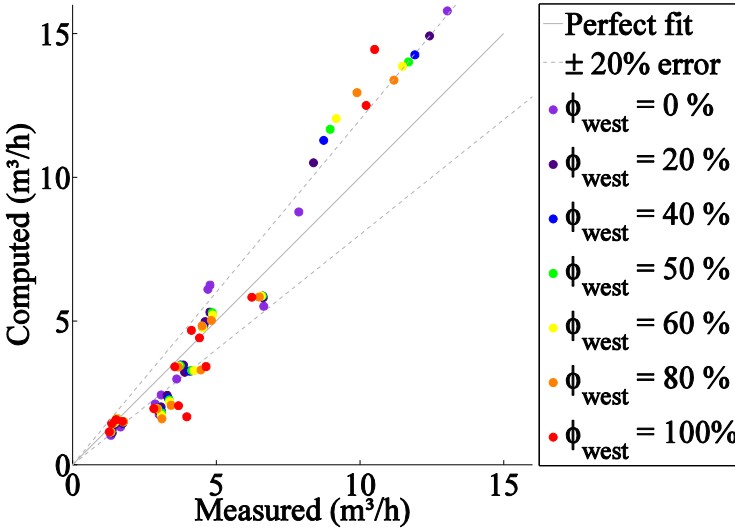

5  Figure 14: Computed vs. observed outflow discharges in each street for different partitions of the inflow discharge between the west and north faces.



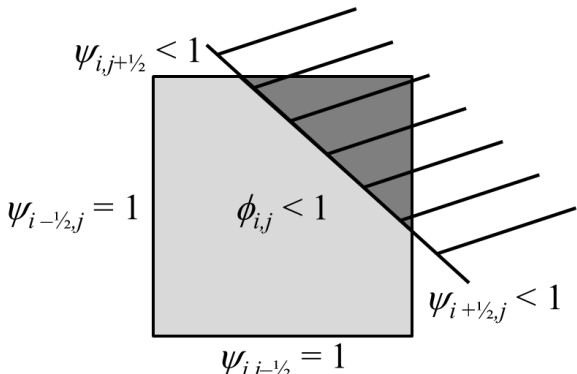

Figure 15: Storage porosity $\phi$ defined for each cell $(i, j)$ of the two-dimensional grid and conveyance porosities $\psi$ defined at the four faces of each grid cell.

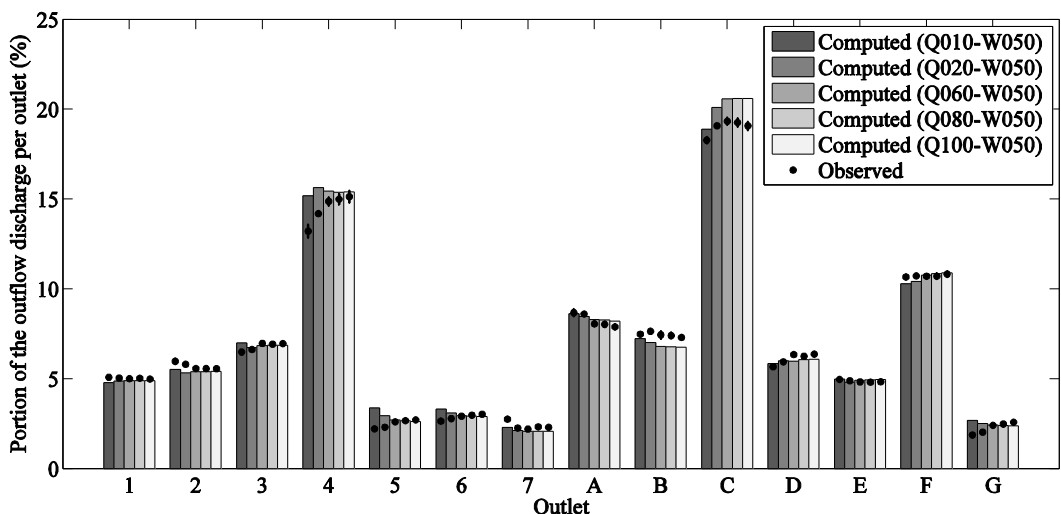

5  Figure 16: Contributions of each street to the total outflow discharge for five different inflow discharges: observations vs. results computed with the shallow-water equations with porosity.