# Peer review of "Hydrodynamics of long-duration urban floods: experiments and numerical modelling"

_Natural Hazards and Earth System Sciences, 2016_

## Short Comment (SC1) · 11 Mar 2016

The paper deals with flood propagation in urban areas. Authors performed laboratory experiments in a typical urban district, containing 7 streets along each direction (in total 49 intersections). In total, 16 tests were conducted (i.e. 16 inflow conditions). The authors apply a 2D shallow-water model to simulate the experimental set-up and investigate the role of roughness and turbulence model. They also discuss the up-scaling of the laboratory observations to the field. The topic of the paper is of interest for NNES readers. The paper is well structured and generally well-written. The laboratory experiments are new and complete the existing ones, although I regret that neither velocity nor flow depth in street intersections were measured, which would provide a nice assessment of the performance of numerical models. Many researchers have provided empirical equations giving the flow partition in street-intersections, but

authors cannot use their laboratory measurements to assess some equations (which are actually useful when 1D models are used for simulating urban floods) because the experimental flow partition between is not avaialble in-between all the intersections. The 2D numerical simulations and comparisons with the laboratory observations are sound. I particularly appreciated the discussion section. I would recommend acceptation of the paper with minor revisions. There are still some issues to be addressed by the authors. The most important are summarized in the document here enclosed

Please also note the supplement to this comment:
http://www.nat-hazards-earth-syst-sci-discuss.net/nhess-2016-10/nhess-2016-10-SC1-supplement.pdf

**Supplement:**

**Review of "Hydrodynamics of long-duration urban floods: experiments and numerical modelling"**

The paper deals with flood propagation in urban areas. Authors performed laboratory experiments in a typical urban district, containing 7 streets along each direction (in total 49 intersections). In total, 16 tests were conducted (i.e. 16 inflow conditions). The authors apply a 2D shallow-water model to simulate the experimental set-up and investigate the role of roughness and turbulence model. They also discuss the up-scaling of the laboratory observations to the field.

The topic of the paper is of interest for NNES readers. The paper is well structured and generally well-written. The laboratory experiments are new and complete the existing ones, although I regret that neither velocity nor flow depth in street intersections were measured, which would provide a nice assessment of the performance of numerical models. Many researchers have provided empirical equations giving the flow partition in street-intersections, but authors cannot use their laboratory measurements to assess some equations (which are actually useful when 1D models are used for simulating urban floods) because the experimental flow partition between is not avaialble in-between all the intersections. The 2D numerical simulations and comparisons with the laboratory observations are sound. I particularly appreciated the discussion section.

I would recommend acceptation of the paper with minor revisions. There are still some issues to be addressed by the authors. The most important are:

**Abstract**

- Please, complete the abstract by giving the most the most important results of the work.
- Specify that the used numerical model is two-dimensional.

**Introduction**

- Water marks are sometimes available but with very high uncertainty. Also, validation of 2D models should be performed on velocity fields, which are not usually available. Authors are invited to comment these two issues (uncertainty and flow velocity fields) in the introduction.
- Authors do not evoke the need of an accurate estimate of the roughness in urban areas (which is, indeed, spatially variable).
- The laboratory work by Paquier et al. (2009) should be also discussed. Finally, the recent work by Bazin (2013) merits to be discussed.

Paquier, A., Tachrift, H., Riviere, N., El kadi Abderrezzak, K. 2009. Assessing the effects of two non-structural flood mitigation measures using laboratory and real cases. Road map towards a flood resilient urban environment, 26/11/2009 - 27/11/2009, Paris, FRA. Proceedings Final conference of the COST action C22. 8p.

Bazin, P.H. 2013. Flows during floods in urban areas: influence of the detailed topography and the exchanges with the sewer system. Phd dissertation, https://hal.inria.fr/tel-01159518/document

- I find the literature review of experimental studies on urban flows too long. It is not necessary to provide the findings of each study. Please, shorten.
- P.4, L8: "..with respect to the main modelling characteristics". Unclear sentence. Please reformulate

**Experiments**

- It is not clear where water profile was measured? In the central axis?. If yes, this limits the interpretation of the results and drawing of conclusions. Why not measuring near the intersections, where the flow is 3D in character.
- Authors are invited to explain why velocity measurements were not performed during the experiment.
- Please explain how the experiment was scaled according to the Froude similarity.
- I think that Colebrook formula was proposed for rough turbulent flow. Setting k = 0 in this formula yields the Von Karman formula. May be replace Colebrook by Von Karman.
- A description of the hydraulic structures (if any) in the street intersections would be interesting. Authors may discuss the observations according to the existing ones (Mignot, Rivière…).
- Authors are invited to explain why the outflow partition remains virtually independent of the total inflow discharge.

**Numerical simulations**
- Specify the time step of the numerical simulation.
- Discrepancy observed in the curved streets are attributed to the Cartesian grid used, which relies on a "staircase" approximation of the obstacles not aligned with the grid. In practice, urban areas are more complex (presence of different obstacles, street angles…) than the "author" experimental set-up. In what extent, the used Cartesian grid (which induces extra flow resistance) can be recommended in field cases?
- Why a particular 2.5 mm grid was tested for the particular total inflow of 20 m³/h?
- Does the grid impact the computation of the hydraulic structures in street intersections?
- Porosity model: is it an isotropic porosity model?
- The used porosity model includes two porosities (storage and conveyance), which is in my opinion sound. However, so such detailed model cannot be easily applied to field cases, because the spatial distribution of porosities is needed. In what way can a model of this type contribute to flood risk studies based on the scale and accuracy at which flow attributes (depth, velocity) are predicted? How the model would be constructed to account for spatial distributions of porosity which might be required for practical applications?

**Conclusions**
- Too long. Please shorten and keep only the most important findings.
- Use only one tense to summarize: either present or past.

---

## Referee Comment (RC1) · Anonymous Referee #1 · 16 Mar 2016

Comments on the paper titled "Hydrodynamics of long-duration urban floods: experiments and numerical modelling" by Anaïs Arrault, Pascal Finaud-Guyot, Pierre Archambeau, Martin Bruwier, Sébastien Erpicum, Michel Pirotton & Benjamin Dewals.

The paper deals with the computation of large urban floods; it is thus perfectly in the scope of the journal. The paper provides firstly an interesting database gathered on a huge laboratory set-up reproducing several streets crossing to form a block. This database is then used to evaluate a 2Dh computational code based on shallow water equations. This code is used to perform a sensitivity analysis to roughness, grid refinement and turbulence models. A discussion addresses the problem of upscaling and the improvements with to the use of a porosity model. The paper is easy to read and well written. It would benefit from addressing some remaining issues. I would recommend the paper for publication after minor revisions.

General comments:

In the present form, the paper is classically written with (i) experiments, (ii) code evaluation and (iii) additional simulations. Yet, this code was already used and evaluated in several studies cited in the reference list. The novelty of the paper, apart from the experiments in a very original facility, would take benefits from addressing more deeply some physical aspects, with the help of the additional flow features provided by the simulations. First, the results are very little influenced by the roughness. But, what was the expected effect of the roughness and why it has no influence? This could be attributed to a predominant role of control sections, but the modification of the recirculation zone when changing the turbulence modelling seems to hardly affect the discharge distribution. This should be commented. The two preceding questions are connected to a last one: is there an explanation to have a 60%-40% downstream distribution instead of 50%-50%? Is it possible to compare this distribution to the one of a single crossroad with the same boundary conditions, using one of the references cited?

Questions and specific comments :

Q1/ Up to 7 authors co-signed the paper. This paper has a very significant experimental part. From references throughout the manuscript and from the acknowledgements, it can be understood that a significant part of these experiments were performed by Araud. Yet, (s)he is not one of the co-authors. This is maybe justified but is mandatory to be checked.

Q2/ Page 3, it is stated that "flash floods" are out of scope of the study. Nevertheless, the time scales associated with present study and flash floods should be detailed (can't it be considered as a succession of steady states), so as the time scale which is considered to discriminate the two kinds of floods. This time scale could be addressed in the section about upscaling, and discussed along the time scales characterizing the laboratory small-scale experiments.

Q3/ About boundary conditions: I understand from the text that the experiments were

conducted with a horizontal bottom but this should be stated more precisely in the manuscript. Could the authors confirm, as it seems to be stated L26-27 page 4, that the linear (q=Q/b) inlet flow rate was constant within all the inlet streets? Finally, are the free-flow conditions performed thanks to chutes?

Q4/ Can some details be provided about the "optical gauge" P5L7? The uncertainty seems quite high (+-1mm) compared to classical devices.

Q5/ I completely agree with authors about the use of a Darcy-Weisbach coefficient instead of a Manning coefficient (p6L11). Nevertheless, the reason could be stated more clearly. Both of these coefficients are "process oriented": one for the channel, one for the pipes. As far as I am concerned, the use of a Darcy-Weisbach coefficient is required here due to the limited values of the Reynolds numbers in the laboratory experiments. The "fully rough regime" is not guaranteed, which prevent from using safely a Manning coefficient depending only on the wall roughness.

Q6/ P7 L8-15 : the downstream discharge distribution is about 60% in the streamwise (inlet) direction and 40% in the crosswise direction. This is not 50%-50% and can some reasons be proposed for this ratio: a slope (but I understand the slope was nil, see Q3), a reference with a single crossroad?

Q7/ P9 L4-8. The location of the water depth profiles drawn should be specified: middle (centreline) of the street, average on a section, . . . Notably, "significant variations" are commented but the comments should account for a possible crossing of the recirculation zones or, instead, of the vena contracta. I expect slightly different comments regarding one case or the other.

Q8/ P10L30 : multiplying the cells by a factor 4 increases the computational cost by 8. Can this be commented?

Q9/ Typical values of the Froude number should be added in table 5

Q10/ The Reynolds number is defined using the water depth, i.e. assuming that the

hydraulic diameter can be assimilated to 4h. This assumption should be valid for pro-
totypes 2 and 3 but is more questionable for the laboratory model and the prototype
1. Was it taken into account to compute the values of the Darcy-Weisbach coefficient
reported in section 5.1?

Typing errors:

- Page 1: the third address in the authors' affiliations is not complete - P6 L25: "a first
test series of tests" - P8 L27-30: What is the "supplement" cited twice? In case it is on
the website of the journal, please do not account for this comment.

Does the paper address relevant scientific and/or technical questions within the scope
of NHESS? (Y) Does the paper present new data and/or novel concepts, ideas, tools,
methods or results? (Y) Are these up to international standards? (Y) Are the scientific
methods and assumptions valid and outlined clearly?  (Not always) Are the results
sufficient to support the interpretations and the conclusions? (Y) Does the author reach
substantial conclusions? (Not always) Is the description of the data used, the methods
used, the experiments and calculations made, and the results obtained sufficiently
complete and accurate to allow their reproduction by fellow scientists (traceability of
results)? (Y) Does the title clearly and unambiguously reflect the contents of the paper?
(Y) Does the abstract provide a concise, complete and unambiguous summary of the
work done and the results obtained?  (Y) Are the title and the abstract pertinent, and
easy to understand to a wide and diversified audience? (Y) Are mathematical formulae,
symbols, abbreviations and units correctly defined and used? If the formulae, symbols
or abbreviations are numerous, are there tables or appendixes listing them?  (Y) Is
the size, quality and readability of each figure adequate to the type and quantity of
data presented?  (Y) Does the author give proper credit to previous and/or related
work, and does he/she indicate clearly his/her own contribution? (See my remark Q1
concerning authors) Are the number and quality of the references appropriate?  (Y)
Are the references accessible by fellow scientists? (Y) Is the overall presentation well
structured, clear and easy to understand by a wide and general audience?  (Y) Is

the length of the paper adequate, too long or too short? (Y) Is there any part of the paper (title, abstract, main text, formulae, symbols, figures and their captions, tables, list of references, appendixes) that needs to be clarified, reduced, added, combined, or eliminated? (N) Is the technical language precise and understandable by fellow scientists? (Y) Is the English language of good quality, fluent, simple and easy to read and understand by a wide and diversified audience? (Y) Is the amount and quality of supplementary material (if any) appropriate? (Y)

---

## Author Comment (AC1) · 27 Apr 2016

We deeply acknowledge the Referees for their detailed analysis of our manuscript and their valuable inputs. We provide as a Supplement a point-by-point response to the main comments by Anonymous Referee #1. The corresponding changes will be implemented in the revised version of the manuscript.

Please also note the supplement to this comment:
http://www.nat-hazards-earth-syst-sci-discuss.net/nhess-2016-10/nhess-2016-10-AC1-supplement.pdf

[Figure]

**Supplement:**

**Author response to Anonymous Referee #1 for "Hydrodynamics of long-duration urban floods: experiments and numerical modelling"**

Anaïs Arrault[1,3], Pascal Finaud-Guyot[2], Pierre Archambeau[1], Martin Bruwier[1], Sébastien Erpicum[1], Michel Pirotton[1] & Benjamin Dewals[1]

[1] Research group HECE, Department ArGEnCo, University of Liege, Liege, Belgium
[2] ICube, Université de Strasbourg, CNRS (UMR 7357), ENGEES, 2 rue Boussingault, Strasbourg, France
[3] Ecole Nationale Supérieure des Mines d'Ales, 6 Avenue de Clavières 30319 Ales Cédex, France

*Correspondence to*: B. Dewals (b.dewals@ulg.ac.be)

We deeply acknowledge the Referees for their detailed analysis of our manuscript and their valuable inputs. We provide hereafter a point-by-point response to the main comments by Anonymous Referee #1. The corresponding changes will be implemented in the revised version of the manuscript.

**General comments**

> The paper deals with the computation of large urban floods; it is thus perfectly in the scope of the journal. The paper provides firstly an interesting database gathered on a huge laboratory set-up reproducing several streets crossing to form a block. This database is then used to evaluate a 2Dh computational code based on shallow water equations. This code is used to perform a sensitivity analysis to roughness, grid refinement and turbulence models. A discussion addresses the problem of upscaling and the improvements with to the use of a porosity model. The paper is easy to read and well written. It would benefit from addressing some remaining issues. **I would recommend the paper for publication after minor revisions.**

Thank you.

> In the present form, the paper is classically written with (i) experiments, (ii) code evaluation and (iii) additional simulations. Yet, this code was already used and evaluated in several studies cited in the reference list. The novelty of the paper, apart from the experiments in a very original facility, would take benefits from addressing more deeply some physical aspects, with the help of the additional flow features provided by the simulations. First, the results are very little influenced by the roughness. But, what was the expected effect of the roughness and why it has no influence? This could be attributed to a predominant role of control sections, but the modification of the recirculation zone when changing the turbulence modelling seems to hardly affect the discharge distribution. This should be commented. The two preceding questions are connected to a last one: is there an explanation to have a 60%-40% downstream distribution instead of 50%-50%? Is it possible to compare this distribution to the one of a single crossroad with the same boundary conditions, using one of the references cited?

The Referee recommends taking more benefit from the additional flow features made accessible by the numerical simulations. In this respect, we undertook several additional simulations and/or analyses of the results, such as:

- Figures 5 and 6 of the original manuscript have been revised to incorporate the computed crosswise variations of the flow depth, which were not available from the experimental measurements (see Figures 1 and 2 of our response to the comments of Anonymous Referee #2);

- new unsteady simulations have been performed to discuss more quantitatively to which extent the experimental observations can be transposed to real-world flood events regarded as a series of successive steady states (see Figure 1 and our response to Specific comment Q2 below);

- we have conducted extra simulations in the simplified configuration of a single "equivalent" intersection (see our response to the Specific comment Q6 below, as well as our response to Anonymous Referee #2), to better elucidate the role of different aspects of the geometry (streets width, location of wide streets, streets inclination) on the partition of the outflow discharge (~ 40 % through the east fact *vs.* ~ 60 % through the south face);

- …

The effects of roughness and control sections are also further discussed in our responses to the specific comments of the Referees.

**2 Specific comments**

Q1/ Up to 7 authors co-signed the paper. This paper has a very significant experimental part. From references throughout the manuscript and from the acknowledgements, it can be understood that a significant part of these experiments were performed by Araud. Yet, (s)he is not one of the co-authors. This is maybe justified but is mandatory to be checked.

As detailed in the Acknowledgement section of the original manuscript, M. Debaucheron helped with numerical simulations and Q. Araud performed experimental measurements; but they both did not take part in the analysis presented in this paper nor contributed to writing the paper. Therefore, we deemed appropriate to acknowledge their contributions.

Q2/ Page 3, it is stated that "flash floods" are out of scope of the study. Nevertheless, the time scales associated with present study and flash floods should be detailed (can't it be considered as a succession of steady states), so as the time scale which is considered to discriminate the two kinds of floods. This time scale could be addressed in the section about upscaling, and discussed along the time scales characterizing the laboratory small-scale experiments.

Flash floods are often associated to short duration and intrinsically transient flood events. This is however not always the case, as stated correctly by the Referee and also, for instance, by Gaume *et al.* (2009):

"*The duration … (of flash) floods depends on the causative storm and hence on the climatic setting. Most generally, the storms inducing flash floods lead to local rainfall accumulations exceeding 100 mm over a few hours … longer lasting stationary storm events may, however, occur in some meteorological contexts, especially in the Mediterranean region.*"

Therefore, we have removed from the Introduction the sentence "*Flash floods are therefore out of the scope of the present study*". Instead, we have elaborated the following discussion.

We undertook extra simulations in unsteady mode, with the purpose of identifying a characteristic time scale of the experimental model. The initial condition corresponds to virtually no water in the model (initial water depth = 1 mm) and no flow. At the initial time, the inflow was suddenly raised to its maximum value upstream of each street. We considered inflow discharges of 20, 60, 80 and 100 m³/h, as well as the following parameters: $\phi_{west} = 50\,\%$, no turbulence model, smooth bottom ($k = 0$ m) and $\Delta x = 1$ cm.

Time series of computed water depths in the centre of the most upstream intersection (i.e. between streets 1 and A) and between the two wide streets 4 and C are displayed in Figure 1. They reveal that the time necessary for reaching a steady state is of the order of $30 \div 60$ s at the scale of the laboratory model.

The scale factor for time is given by the ratio between the scale factor for horizontal lengths ($1 / e_H$) and the scale factor for velocity. The latter is the square root of the scale factor for vertical dimensions ($1 / e_V$), consistently with the Froude similarity adopted here. Hence, the characteristic times obtained at the scale of the laboratory model must be magnified by $e_H / (e_V)^{0.5}$ to obtain the corresponding characteristic times at the prototype scale.

- For Prototype 1, this leads to a magnification factor of $200 / 200^{0.5}$, hence to a characteristic time of the order of $7 \div 14$ min.
- For the more realistic Prototypes 2 and 3, a magnification factor of $200 / 20^{0.5}$ is obtained, which leads to a characteristic time of the order of $20 \div 45$ min.

In conclusion, the observations of the present research remain valid provided that the considered flood events remain sufficiently gradual, i.e. that the characteristic time scales of the flood waves remain above $20 \div 45$ min. This will be explained in the revised manuscript and, by the way, is consistent with the title of the manuscript ("long-duration urban floods").
* * *
Q3/ About boundary conditions: I understand from the text that the experiments were conducted with a horizontal bottom but this should be stated more precisely in the manuscript. Could the authors confirm, as it seems to be stated L26-27 page 4, that the linear (q=Q/b) inlet flow rate was constant within all the inlet streets? Finally, are the free-flow conditions performed thanks to chutes?
* * *
The use of a horizontal bottom will be clearly stated at the beginning of section 2.1.1 in the revised manuscript.

We also confirm that the specific discharge (i.e. discharge per unit width of the street at inlet) was the same at the inlet of all streets of a given face (west or north). The specific discharge differs from one face to the other because the total inlet width

is different (one wide street along the west face *vs.* two wide streets along the north face). Moreover, in tests with $\phi_{west} \neq 50$ %, the specific discharge obviously differs from one face to the other.

The free-flow conditions at the outlet of each street are realized experimentally thanks to chutes, as shown in Figure 2.

> Q4/ Can some details be provided about the "optical gauge" P5L7? The uncertainty seems quite high (+-1mm) compared to classical devices.

Water level measurements were conducted using an optical gauge fixed on an automatic traverse system. The gauge detects the phase (air *vs.* water) in which it is located. The measurement uncertainty results therefore mainly from the accuracy of the motor, which was estimated at ± 1 mm. This will be explained in the revised version of the manuscript (section 2.1.2). This optical device was preferred here compared to more standard acoustic techniques due to the narrow character of the streets which would have induced reflections of the sound waves on the walls and disruptions in the measurements if an acoustic device had been used.

> Q5/ I completely agree with authors about the use of a Darcy-Weisbach coefficient instead of a Manning coefficient (p6L11). Nevertheless, the reason could be stated more clearly. Both of these coefficients are "process oriented": one for the channel, one for the pipes. As far as I am concerned, the use of a Darcy-Weisbach coefficient is required here due to the limited values of the Reynolds numbers in the laboratory experiments. The "fully rough regime" is not guaranteed, which prevent from using safely a Manning coefficient depending only on the wall roughness.

The Referee is right. This will be stated as follows in section 2.2 of the revised manuscript:

"*Also, the experimental conditions do not a guarantee a hydraulic rough flow regime, which is necessary for applying Manning formula.*"

> Q6/ P7 L8-15 : the downstream discharge distribution is about 60% in the streamwise (inlet) direction and 40% in the crosswise direction. This is not 50%-50% and can some reasons be proposed for this ratio: a slope (but I understand the slope was nil, see Q3), a reference with a single crossroad?

The bottom is indeed flat, as will be mentioned in section 2.1.1 of the revised manuscript. Nonetheless, for $\phi_{west} = 50$ %, the experimental observations indicate that the outflow through the east face is about 40 % of the total inflow, while the outflow through the south face is 60 % of the total inflow. The numerical computations also confirm these observations (see Figure 2 of the original manuscript).

Surely, this is partly explained by the total flow width available along the north-south direction compared to the west-east direction. Indeed, only one "wide" street (street 4) is aligned along the west-east direction, whereas two of them (streets C and F) convey the flow in the north-south direction. As a result, the total flow width in the west-east direction is 42.5 cm,

which is lower than the total flow width along the north-south direction (50 cm). Consequently, the available flow width along the east face is 46 % of the total outflow width, whereas it is 54 % for the south face. This difference goes in the same direction as the difference in outflows (40 % *vs.* 60 %).

[See also our response to Anonymous Referee #2.]

5  To test this hypothesis of "attraction" effect of the wider streets, we undertook additional simulations corresponding to a single "equivalent" 4-branch intersection, with the north-south and west-east streets widths respectively equal to 0.5 m and 0.425 m. These widths mimic the cumulative street widths along the north-south and the west-east directions in the experimental model (respectively equal to 0.5 m and 0.425 m). We performed the simulations for the two extreme discharges (20 m$^3$/h and 100 m$^3$/h), with equal inflow partition between the west and north faces ($\phi_{west} = 50$ %) and assuming a smooth

10  bottom ($k = 0$ m). Free flow boundary conditions were prescribed at the downstream end of each street, located at a long distance downstream of the crossroad (8 times the street width). We used the finest grid spacing considered in the paper ($\Delta x = 2.5$ mm) and we tested the computations with and without activation of the turbulence model.

For both inflow discharges (20 m$^3$/h and 100 m$^3$/h), the computed results reveal a partition of the outflow discharge proportional to the street widths (54 % *vs.* 46 %). The same results were obtained with and without activation of the

15  turbulence model. Nonetheless, this geometric effect explains only partly the difference in the experimentally observed outflow discharges (60 % *vs.* 40 %). We attribute the remaining difference to features which are not properly reflected in the single "equivalent" intersection considered here (e.g., the spatial distribution of the wider streets within the scale model); but which are expected to further amplify the difference in the outflow discharges between the north-south and the west-east directions. Particularly, the downstream parts of the streets aligned along the west-east direction (streets 1 to 7) are all

20  inclined towards the north, i.e. towards upstream as far as the north-south direction is concerned. This surely contributes to further reduce the outflow through the east face. This is will be explained in section 3.1.1 of the revised manuscript.
* * *
Q7/ P9 L4-8. The location of the water depth profiles drawn should be specified: middle (centreline) of the street, average on a section, : : : Notably, "significant variations" are commented but the comments should account for a possible crossing of the recirculation zones or, instead, of the vena contracta. I expect slightly different comments regarding one case or the other.
* * *
25  The water depth profiles are drawn along the centreline of the streets because experimental data have not been collected elsewhere. This will be explicitly stated in section 3.1.4 of the revised manuscript. For the sake of consistency, the displayed computed water depths were also taken along the centreline of the streets.

So far, water depths have not been measured beyond the streets centreline because the experimental procedure for conducting water level measurements was particularly slow. For each test, water levels were measured at about 600 locations

30  along the centreline of the streets using the automatic traverse system (see also Q. 4). A single survey of this type (600 points) took almost one whole day. This will be detailed in section 2.1.2 of the revised manuscript. In the future, more detailed water level measurements will be performed in the near-field of the street intersections.

In section 3.1.4, the wording "most significant variations" refers to variations in the *streamwise* direction. This will be clarified in the text of the revised manuscript. However, we have no experimental information available to state whether the observed profiles along centreline of the streets cross recirculations and/or the *vena contracta* in the experiments.

As detailed in our response to Anonymous Referee #2, we have revised Figures 5 and 6 of the original manuscript to examine the crosswise distribution of water depths in the numerical results. This is displayed by the shaded area (■) in Figures 1 and 2 of our response to Anonymous Referee #2. These additional data confirm that significant crosswise variations in the water depths are located immediately downstream of the street intersections, which is consistent with the location of recirculation zones and *vena contracta*.

Q8/ P10L30 : multiplying the cells by a factor 4 increases the computational cost by 8. Can this be commented?

This results from the *Courant-Friedrichs-Lewy* (CFL) condition for explicit time integration (e.g., Bates et al., 2010), which states that the time step $\Delta t$ must scale with the space step $\Delta x$ to preserve the stability of the numerical scheme: $\Delta t \sim \Delta x$. Hence, when the grid spacing $\Delta x$ is reduced by a factor two, the number of cells in 2D increases by a factor four and the time step is reduced by a factor two. As a result, the overall computational cost increases by a factor eight. This will be detailed in the revised version of the manuscript.

Q9/ Typical values of the Froude number should be added in table 5

The typical values of the Froude number remain the same for the laboratory model and for the three prototypes, since the Froude similarity was used for upscaling the experimental results in all cases (whether distorted or non-distorted). Therefore, we prefer not to include the Froude number in Table 5; but to report its typical values (0.15 - 0.4 close to the inlets) in the main text (at the end of section 2.1.3, Test program) of the revised manuscript.

Q10/ The Reynolds number is defined using the water depth, i.e. assuming that the hydraulic diameter can be assimilated to 4h. This assumption should be valid for prototypes 2 and 3 but is more questionable for the laboratory model and the prototype 1. Was it taken into account to compute the values of the Darcy-Weisbach coefficient reported in section 5.1?

The Referee is right that the hydraulic radius may be much smaller than the water depth in the laboratory model and in Prototype 1, particularly in the case of high inflow discharge in the narrow streets. However, the definition of the Reynolds number used in the manuscript ($R = 4\,h\,u\,/\,\nu$) is consistent with the standard formulation used in 2D-horizontal flow models, while the general definition based on the hydraulic radius ($R_{1D} = 4\,R_h\,u\,/\,\nu$, with $R_h = b\,h\,/\,(\,b + 2\,h\,)$ and $b$ = street width) is mostly used in the context of 1D flow modelling.

Nonetheless, we took the Referee's remark into consideration and we display below a modified version of Table 5, in which we also include the values obtained by using $R_{1D}$ instead of $R$. By the way, we also corrected the values of $R$ for a missing

factor 1/2 (in the original manuscript, we evaluated $R$ in Table 5 considering that the *total* inflow was supplied to *each* face, which is obviously not the case).

In the end, the only consequences of using the "water depth-based" Reynolds number $R$ instead of the "hydraulic radius-based" Reynolds number $R_{1D}$, are the following:

- the range of $f$ is $2 \times 10^{-2} \div \underline{3} \times 10^{-2}$, instead of $2 \times 10^{-2} \div \underline{4} \times 10^{-2}$ for the laboratory model,
- $f$ is estimated equal to ~ $2 \times 10^{-2}$, instead of being in the range $2 \times 10^{-2} \div \underline{3} \times 10^{-2}$ for prototype 1,
- and the range of $f$ is $4 \times 10^{-2} \div \underline{7} \times 10^{-2}$, instead of $4 \times 10^{-2} \div \underline{6} \times 10^{-2}$ for prototype 3.

Those slight variations in the values of $f$ do not result in any significant change in the discussion of section 5.1 in the manuscript. Therefore, our suggestion is not to modify the definition of the Reynolds number in the manuscript as this would

10 not add to the main message we want to convey.

**3 Typing errors**

- Page 1: the third address in the authors' affiliations is not complete
- P6 L25: "a first test series of tests"

This will be corrected in the revised manuscript.

15 - P8 L27-30: What is the "supplement" cited twice? In case it is on the website of the journal, please do not account for this comment.

Supplements are indeed available on the website of the journal.

**References cited here but not in the original manuscript**

Gaume, E., Bain, V., Bernardara, P., Newinger, O., Barbuc, M., Bateman, A., Blaškovičová, L., Blöschl, G., Borga, M.,
20 Dumitrescu, A., Daliakopoulos, I., Garcia, J., Irimescu, A., Kohnova, S., Koutroulis, A., Marchi, L., Matreata, S., Medina, V., Preciso, E., Sempere-Torres, D., Stancalie, G., Szolgay, J., Tsanis, I., Velasco, D., Viglione, A. (2009). A compilation of data on European flash floods. *Journal of Hydrology*, **367**(1-2), 70-78.

Bates, P.D., Horritt, M.S., Fewtrell, T.J. (2010). A simple inertial formulation of the shallow water equations for efficient two-dimensional flood inundation modelling. *Journal of Hydrology*, **387** (1-2), 33-45.

**Tables**

Table 1 (expanded version of Table 5 in the original manuscript): Characteristic Reynolds numbers R and $R_{1D}$, roughness height $k$ and corresponding Darcy-Weisbach coefficient $f$ at the laboratory model scale and at the prototype scale (real-world) as a function of the horizontal and vertical magnification factors $e_H$ and $e_V$.

| | Laboratory model | Prototype 1 | Prototype 2 | Prototype 3 |
|---|---|---|---|---|
| |  ~10 cm ~10 cm |  ~20 m ~20 m |  ~2 m ~20 m |  ~2 m ~20 m |
| $e_H$ | - | 200 | 200 | 200 |
| $e_V$ | - | 200 | 20 | 20 |
| $k_s$ | $< 10^{-5}$ m | $\sim 5 \times 10^{-2}$ m | $\sim 5 \times 10^{-3}$ m | $\sim 0.1$ m |

Reynolds number based on the water depth: $R = 4\,h\,u\,/\,\nu$

| | Laboratory model | Prototype 1 | Prototype 2 | Prototype 3 |
|---|---|---|---|---|
| R | $1 \times 10^4 \div 1 \times 10^5$ | $3 \times 10^7 \div 4 \times 10^8$ | $1 \times 10^6 \div 1 \times 10^7$ | $1 \times 10^6 \div 1 \times 10^7$ |
| $k\,/\,4\,h$ | $3 \times 10^{-5} \div 7 \times 10^{-5}$ | $7 \times 10^{-4} \div 2 \times 10^{-3}$ | $7 \times 10^{-4} \div 2 \times 10^{-3}$ | $1 \times 10^{-2} \div 4 \times 10^{-2}$ |
| $f$ | $2 \times 10^{-2} \div 3 \times 10^{-2}$ | $\sim 2 \times 10^{-2}$ | $\sim 2 \times 10^{-2}$ | $4 \times 10^{-2} \div 6 \times 10^{-2}$ |

Reynolds number based on the hydraulic radius : $R_{1D} = 4\,R_h\,u\,/\,\nu$, with $R_h = b\,h\,/\,(\,b + 2\,h\,)$ and $b$ = street width

| | Laboratory model | Prototype 1 | Prototype 2 | Prototype 3 |
|---|---|---|---|---|
| $R_{1D}$ | $5 \times 10^3 \div 5 \times 10^4$ | $1 \times 10^7 \div 2 \times 10^8$ | $9 \times 10^5 \div 1 \times 10^7$ | $9 \times 10^5 \div 1 \times 10^7$ |
| $k\,/\,4\,R_h$ | $7 \times 10^{-5} \div 2 \times 10^{-4}$ | $2 \times 10^{-3} \div 4 \times 10^{-3}$ | $8 \times 10^{-4} \div 2 \times 10^{-3}$ | $2 \times 10^{-2} \div 4 \times 10^{-2}$ |
| $f$ | $2 \times 10^{-2} \div 4 \times 10^{-2}$ | $2 \times 10^{-2} \div 3 \times 10^{-2}$ | $\sim 2 \times 10^{-2}$ | $4 \times 10^{-2} \div 7 \times 10^{-2}$ |

**Figures**

[Figure]

(a)

(b)

Figure 1: Computed water depths in the unsteady simulations, at the intersections (a) between streets 1 and A, and (b) between streets 4 and C. The markers (•) indicate the moment when the water depth reaches 99 % of its ultimate value.

[Figure]

Figure 2: Outlet of a street in the experimental model.

---

## Author Comment (AC2) · 27 Apr 2016

We deeply acknowledge the Referees for their detailed analysis of our manuscript and their valuable inputs. We provide as a Supplement a point-by-point response to the main comments by Anonymous Referee #2. The corresponding changes will be implemented in the revised version of the manuscript.

Please also note the supplement to this comment:
http://www.nat-hazards-earth-syst-sci-discuss.net/nhess-2016-10/nhess-2016-10-AC2-supplement.pdf

[Figure]

**Supplement:**

**Author response to Anonymous Referee #2 for "Hydrodynamics of long-duration urban floods: experiments and numerical modelling"**

Anaïs Arrault[1,3], Pascal Finaud-Guyot[2], Pierre Archambeau[1], Martin Bruwier[1], Sébastien Erpicum[1], Michel Pirotton[1] & Benjamin Dewals[1]

[1] Research group HECE, Department ArGEnCo, University of Liege, Liege, Belgium
[2] ICube, Université de Strasbourg, CNRS (UMR 7357), ENGEES, 2 rue Boussingault, Strasbourg, France
[3] Ecole Nationale Supérieure des Mines d'Ales, 6 Avenue de Clavières 30319 Ales Cédex, France

*Correspondence to*: B. Dewals (b.dewals@ulg.ac.be)

We deeply acknowledge the Referees for their detailed analysis of our manuscript and their valuable inputs. We provide hereafter a point-by-point response to the main comments by Anonymous Referee #2. The corresponding changes will be implemented in the revised version of the manuscript.

**1 General comments**

> The paper deals with flood propagation in urban areas. Authors performed laboratory experiments in a typical urban district, containing 7 streets along each direction (in total 49 intersections). In total, 16 tests were conducted (i.e. 16 inflow conditions). The authors apply a 2D shallow-water model to simulate the experimental set-up and investigate the role of roughness and turbulence model. They also discuss the up-scaling of the laboratory observations to the field.
>
> The topic of the paper is of interest for NNES readers. The paper is well structured and generally well-written. The laboratory experiments are new and complete the existing ones, although I regret that neither velocity nor flow depth in street intersections were measured, which would provide a nice assessment of the performance of numerical models. Many researchers have provided empirical equations giving the flow partition in street-intersections, but authors cannot use their laboratory measurements to assess some equations (which are actually useful when 1D models are used for simulating urban floods) because the experimental flow partition between is not avaialble in-between all the intersections. The 2D numerical simulations and comparisons with the laboratory observations are sound. I particularly appreciated the discussion section.
>
> **I would recommend acceptation of the paper with minor revisions**. There are still some issues to be addressed by the authors. The most important are: …

In our response to the specific comments below, we details why flow velocity and water depths in the intersections could not be measured so far. Thank you for recommending publication after minor revisions.

**2 Specific comments**

**2.1 Abstract**

> - Please, complete the abstract by giving the most the most important results of the work.
>
> - Specify that the used numerical model is two-dimensional.

5   This will be implemented in the revised version of the manuscript.

**2.2 Introduction**

> Water marks are sometimes available but with very high uncertainty. Also, validation of 2D models should be performed on velocity fields, which are not usually available. Authors are invited to comment these two issues (uncertainty and flow velocity fields) in the introduction.

10   The following sentence will be included in the Introduction section of the revised manuscript:

*"Water marks and aerial imagery provide some relevant information but they remain affected by high uncertainties. They are also far from sufficient to reflect the whole complexity of inundation flows in densely urbanized floodplains, the proper description of which requires information on the velocity fields and discharge partitions."*

> - Authors do not evoke the need of an accurate estimate of the roughness in urban areas (which is, indeed, spatially variable).

15   The following sentence will be introduced in the Introduction of the revised manuscript:

"*Difficulties remain nonetheless for estimating the roughness parameters which may vary significantly in space, particularly in floodplains.*"

> - The laboratory work by Paquier et al. (2009) should be also discussed. Finally, the recent work by Bazin (2013) merits to be discussed.
>
> Paquier, A., Tachrift, H., Riviere, N., El kadi Abderrezak, K. 2009. Assessing the effects of two non-structural flood mitigation measures using laboratory and real cases. Road map towards a flood resilient urban environment, 26/11/2009 - 27/11/2009, Paris, FRA. Proceedings Final conference of the COST action C22. 8p.
>
> Bazin, P.H. 2013. Flows during floods in urban areas: influence of the detailed topography and the exchanges with the sewer system. Phd dissertation, https://hal.inria.fr/tel-01159518/document

25   The contribution of Paquier et al. (2009) will be discussed in the Introduction of the revised manuscript and cited in the reference list. Similarly, the study by Bazin (2013) will be cited at different occasions throughout the revised manuscript, particularly in the Introduction. Part of the work of Bazin (2013) was actually already reported in the original manuscript

(see reference Mignot et al. 2013). In the revised manuscript, we will also refer to the peer-reviewed paper by Bazin et al. (2014).
* * *
- I find the literature review of experimental studies on urban flows too long. It is not necessary to provide the findings of each study. Please, shorten.
* * *
5  It will be shortened in the revised manuscript.
* * *
- P.4, L8: "..with respect to the main modelling characteristics". Unclear sentence. Please reformulate
* * *
This part of the text will be rephrased as follows:

*"The influence of varying the total inflow discharge is analysed in Sect. 3, together with a sensitivity analysis of the computed results considering different roughness parameters, grid spacing and turbulence models."*

10  **2.3 Experiments**
* * *
- It is not clear where water profile was measured? In the central axis?. If yes, this limits the interpretation of the results and drawing of conclusions. Why not measuring near the intersections, where the flow is 3D in character.
* * *
[see also our response to Q. 7 of Anonymous Referee #1]

The water profiles were measured along the centreline of each street. This will be explicitly stated in section 3.1.4 of the
15  revised manuscript. For this reason, comparisons with the numerical simulations were performed by retrieving the computed water depths along the centreline of the streets in the numerical model.

So far, water depths have not been measured beyond the streets centreline because the experimental procedure for conducting water level measurements was particularly slow (optical gauge moving on an automatic traverse system). For each test, water levels were measured at about 600 locations along the centreline of the streets using the automatic traverse
20  system. A single survey of this type (600 points) took almost one whole day. In the future more detailed water level measurements will be performed in the near-field of the street intersections. We will explicitly refer to this as a perspective in the Conclusion of the revised manuscript.

Nonetheless, we used the numerical simulations to investigate to which extent the flow depth varies within a cross-section. As shown by the shaded area (■) in Figures 1 and 2, significant crosswise variations in the flow depths were obtained only
25  in the near-field of the street intersections, which is consistent with the presence of flow structures such as recirculation zones in these areas.

- Authors are invited to explain why velocity measurements were not performed during the experiment.

The measurement devices installed in the laboratory are moved by means of a traverse system, the automatic control of which requires extensive and accurate data, as well as a complex calibration procedure to avoid any contact with the walls and obstacles. Due to the time allocated to this development, velocity measurements could not be performed up to now.

5 In the revised manuscript, we will highlight the need for velocity measurements in the Introduction and mention it also as a perspective at the end of the Conclusion.

- Please explain how the experiment was scaled according to the Froude similarity.

Assuming a scale factor 1 / 200 for horizontal dimensions, the widths of the narrow and wide streets in the laboratory model correspond respectively to 10 m and 25 m at the prototype scale, which is considered as realistic.

10 Using the same scale factor for vertical dimensions would have led to very small water depths in the laboratory model (e.g., a water depth of 2 m in a real-world floodplain would have been represented by a 1 cm water depth in the laboratory model). Such small water depths would have resulted in significant measurement errors and particularly low Reynolds numbers in the laboratory model. Therefore, a distorted model was considered, assuming a vertical scale factor of 1 / 20.

According to the Froude similarity, the scale factor for velocity was defined as the square root of the scale factor for vertical

15 dimensions: $( 1 / 20 )^{0.5} \approx 0.22$. Hence, the scale factor for discharge is: $( 1 / 200 ) \times ( 1 / 20 ) \times ( 1 / 20 )^{0.5} = 5.6 \times 10^{-5}$.

Next, the values of inflow discharge to be supplied to the laboratory model were deduced from typical real-world observations (Mignot et al., 2006) considering moderate and extreme flood conditions, as detailed in Table 1. In the end, the range of investigated inflow discharges was slightly extended to 10 ÷ 100 m³/h.

We believe that details on the scaling of the laboratory model are not a key part of the main message we want to convey in

20 the paper. Therefore, we suggest not inserting this information in the main text; but instead including it as a Supplement of the revised manuscript.

- I think that Colebrook formula was proposed for rough turbulent flow. Setting k = 0 in this formula yields the Von Karman formula. May be replace Colebrook by Von Karman.

We will refer to "von Kármán formula" in the revised manuscript. Nonetheless, we also keep the wording "Colebrook

25 formula" in the text since non-zero values of $k$ are considered in sections 3.2.1 and 5.1.

- A description of the hydraulic structures (if any) in the street intersections would be interesting. Authors may discuss the observations according to the existing ones (Mignot, Rivière…).

In the experiments, flow recirculations could be observed downstream of several crossroads. This is consistent with flow descriptions available in literature (e.g., Weber et al., 2001; Neary et al., 1999). These recirculations and the associated *vena*

5   *contracta* induce significant crosswise variations in the water depths (see also the shaded area in the revised version of Figures 5 and 6). The numerical simulations also highlight the presence of flow recirculations, as shown in Figures 10 and 11 in the original manuscript. Figures 9 and 11 additionally show the presence of control sections in the *vena contracta*, which is consistent with e.g. Figure 5 in Riviere et al. (2014).

In contrast, other hydraulic structures such as hydraulic jumps could hardly be detected in the experiments; but this may

10   result from the absence of velocity measurements. It may also be explained by the overall configuration of the model, in which the downstream intersections generate backwater effects in the upstream region, leading to subcritical flow virtually everywhere.

- Authors are invited to explain why the outflow partition remains virtually independent of the total inflow discharge.

This is an experimental finding which is also confirmed by the 2D numerical simulations. The physical reason has not yet

15   been totally clarified.

One attempt was to use Eq. (14) in Riviere et al. (2011) to show that, at a single intersection, the ratio between one outflow discharge and one inflow discharge (noted $R_q$ in the cited reference) is almost independent of the inflow discharge (expressed by the non-dimensional parameter $R_g$ in the cited reference). Doing so confirms that indeed $R_q$ varies very little with $R_g$ over the range of values of $R_g$ considered here. However, we believe that this is not truly valid because of the significant

20   differences between the geometric and flow conditions considered in the analysis of Riviere et al. (2011) and those of the present experiments. Indeed, in Riviere et al. (2011),

- all four branches are identical (same width), while here streets of different widths are considered;
- the intersections have right angles, which is not the case here;
- the ratio $R_g = Q_i / [ b^2 ( g b )^{0.5} ]$ (with $Q_i$ the inflow discharge in a given street and $b$ the street width) was varied in the

25      range 0.0226 ÷ 0.0651, while in the experiments considered here it varies between $2 \times 10^{-2}$ and $9.3 \times 10^{-1}$.

In a second attempt, we undertook additional simulations corresponding to a single "equivalent" 4-branch intersection, with the north-south and west-east streets widths respectively equal to 0.5 m and 0.425 m (see also our response to Q. 6 of Anonymous Referee #1). These widths mimic the cumulative street widths along the north-south and the west-east directions in the experimental model (respectively equal to 0.5 m and 0.425 m). We performed the simulations for the two extreme

30   discharges (20 m³/h and 100 m³/h), with equal inflow partition between the west and north faces ($\phi_{west}$ = 50 %) and assuming

a smooth bottom ($k = 0$ m). Free flow boundary conditions were prescribed at the downstream end of each street, located at a long distance downstream of the crossroad (8 times the street width). We used the finest grid spacing considered in the paper ($\Delta x = 2.5$ mm) and we tested the computations with and without activation of the turbulence model.

For both inflow discharges (20 m³/h and 100 m³/h), the computed results reveal a partition of the outflow discharge proportional to the street widths (54 % *vs.* 46 %). The same results were obtained with and without activation of the turbulence model. It seems therefore that the geometric effect resulting from the difference in the cumulative street widths along the north-south and west-east directions acts similarly over the whole range of considered inflow discharges (20 m³/h and 100 m³/h).

Nonetheless, this geometric effect explains only partly the difference in the experimentally observed outflow discharges (60 % *vs.* 40 %). We attribute the remaining difference to *(i)* the spatial distribution of the wider streets within the scale model, and *(ii)* the inclination of several streets. These two effects are not properly reflected in the single "equivalent" intersection considered here; but they are expected to further amplify the difference in the outflow discharges between the north-south and the west-east directions.

**2.4 Numerical simulations**

- Specify the time step of the numerical simulation.

The following information will be provided at the end of section 2.2 in the revised manuscript:

"*The time step used in the computations is optimized based on the Courant-Friedrichs-Lewy (CFL) stability condition (e.g., Bates et al., 2010). It takes values of the order of $10^{-3}$ s for simulations of the laboratory model and $5 \times 10^{-2}$ s for the prototype scale.*"

- Discrepancy observed in the curved streets are attributed to the Cartesian grid used, which relies on a "staircase" approximation of the obstacles not aligned with the grid. In practice, urban areas are more complex (presence of different obstacles, street angles…) than the "author" experimental set-up. In what extent, the used Cartesian grid (which induces extra flow resistance) can be recommended in field cases?

The model based on anisotropic porosity parameters (section 5.2) is certainly a viable approach for practical applications. For the experiments considered here, all porosity parameters were *deduced* directly from geometric data and there was no calibration of these porosity parameters: $\phi$ is simply the void fraction in the cell, while $\psi$ is given by the fraction of each cell interface which is not blocked by obstacles. The same approach may apply for real-world cases, for which a digital terrain model (DTM) is used to describe the topography and vector data are available to locate the position of the buildings. Among others, Schubert and Sanders (2012) applied such a technique to simulate the Baldwin Hills urban dam break scenario (see

their "building porosity" approach). Sanders et al. (2008) applied a similar model to the Toce Valley flash flood (see their approach based on "gap-conforming" mesh). This will be mentioned in the conclusion of the revised manuscript.

> - Why a particular 2.5 mm grid was tested for the particular total inflow of 20 m³/h?

When the grid spacing is reduced to 2.5 mm, the computational cost is magnified by a factor ~ 8 compared to a grid spacing
5 of 5 mm and is about 64 higher than with a grid spacing of 1 cm. This is the reason why the simulations with a grid spacing of 2.5 mm were not repeated for all inflow discharges.

> - Does the grid impact the computation of the hydraulic structures in street intersections?

To some extent, it does. This was already acknowledged in the original manuscript. Indeed, in section 3.2.2, we wrote: "*Changing the grid size leads also to local changes in the flow pattern. As an example, Figure 9 shows the details of the flow*
10 *field near the downstream end of street 4 in the case of test Q100-W050 computed with cell sizes of 1 cm and 5 mm … lead in some cases to the development of flow structures (such as cross waves), which as a matter of fact are mesh-dependent (Figure 9a and b). Their impact remains however very limited ...*".

> - Porosity model: is it an isotropic porosity model?

The porosity model is based on anisotropic porosity parameters, as shown in Figure 15 of the original manuscript. This will
15 also be explicitly stated in section 5.2 of the revised manuscript.

> - The used porosity model includes two porosities (storage and conveyance), which is in my opinion sound. However, so such detailed model cannot be easily applied to field cases, because the spatial distribution of porosities is needed. In what way can a model of this type contribute to flood risk studies based on the scale and accuracy at which flow attributes (depth, velocity) are predicted? How the model would be constructed to account for spatial distributions of porosity which might be
> 20 required for practical applications?

[SAME RESPONSE AS FOUR COMMENTS ABOVE]
The model based on anisotropic porosity parameters (section 5.2) is certainly a viable approach for practical applications. For the experiments considered here, all porosity parameters were *deduced* directly from geometric data and there was no calibration of these porosity parameters: $\phi$ is simply the void fraction in the cell, while $\psi$ is given by the fraction of each cell
25 interface which is not blocked by obstacles. The same approach may apply for real-world cases, for which a digital terrain model (DTM) is used to describe the topography and vector data are available to locate the position of the buildings. Among others, Schubert and Sanders (2012) applied such a technique to simulate the Baldwin Hills urban dam break scenario (see

their "building porosity" approach). Sanders et al. (2008) applied a similar model to the Toce Valley flash flood (see their approach based on "gap-conforming" mesh). This will be mentioned in the conclusion of the revised manuscript.

**2.5 Conclusions**

> - Too long. Please shorten and keep only the most important findings.
> - Use only one tense to summarize: either present or past.

This will be corrected in the revised version of the manuscript.

**References**

The cited references which are already listed in the original manuscript are not repeated here.

10    Bazin, P.-H., Nakagawa, H., Kawaike, K., Paquier, A., Mignot, E. (2014). Modeling flow exchanges between a street and an underground drainage pipe during urban floods. *Journal of Hydraulic Engineering*, **140**(10), 04014051.

Neary, V., Sotiropoulos, F., and Odgaard, A. (1999). Three-Dimensional Numerical Model of Lateral-Intake Inflows. *J. Hydraul. Eng.*, **125**(2), 126-140.

**Tables**

15    Table 1: Scaling of the experimental model according to the Froude similarity for moderate and extreme flood conditions as reported by Mignot et al. (2006).

|  | Moderate flood conditions | Extreme flood conditions |
|---|---|---|
| Typical real-world water depth | 0.3 m |  |
| Typical real-world flow velocity | 1 m/s |  |
| Discharge in real-world narrow streets (10 m in width) | 3 m³/s | 20 m³/s |
| Discharge in real-world wide streets (25 m in width) | 7.5 m³/s | 50 m³/s |
| Inflow discharge in the narrow streets of the laboratory model | 0.6 m³/h | 4 m³/h |
| Inflow discharge in the wide streets of the laboratory model | 1.5 m³/h | 10 m³/h |
| Total inflow into the laboratory model | 11.2 m³/h | 74.5 m³/h |

**Figures**

[Figure]

Figure 1 (revised version of Figure 5 in the original manuscript): Observed and computed water depths h for inflow discharges varying between 20 m³/h and 100 m³/h in street C. The shaded area (■) represents the range of variation along the crosswise direction of the computed water depths ($\Delta x$ = 1 cm, with $k$-$\varepsilon$ model).

[Figure]

Figure 2 (revised version of Figure 6 in the original manuscript): Observed and computed water depths h for inflow discharges varying between 20 m³/h and 100 m³/h in street 4. The shaded area (■) represents the range of variation along the crosswise direction of the computed water depths ($\Delta x$ = 1 cm, with $k$-$\varepsilon$ model).